# Action-Minimization Meets Generative Modeling: Efficient Transition Path Sampling with the Onsager-Machlup Functional

**Sanjeev Raja** [*1] **Martin Šípka** [*23] **Michael Psenka** [*1] **Tobias Kreiman** [1] **Michal Pavelka** [2] **Aditi S. Krishnapriyan** [145]

## Abstract

Transition path sampling (TPS), which involves finding probable paths connecting two points on an energy landscape, remains a challenge due to the complexity of real-world atomistic systems. Current machine learning approaches use expensive, task-specific, and data-free training procedures, limiting their ability to benefit from recent advances in atomistic machine learning, such as high-quality datasets and large-scale pre-trained models. In this work, we address TPS by interpreting candidate paths as trajectories sampled from stochastic dynamics induced by the learned score function of pre-trained generative models, specifically denoising diffusion and flow matching. Under these dynamics, finding high-likelihood transition paths becomes equivalent to minimizing the Onsager-Machlup (OM) action functional. This enables us to repurpose pre-trained generative models for TPS in a zero-shot manner, in contrast with bespoke, task-specific TPS models trained in previous work. We demonstrate our approach on varied molecular systems, obtaining diverse, physically realistic transition pathways and generalizing beyond the pre-trained model's original training dataset. Our method can be easily incorporated into new generative models, making it practically relevant as models continue to scale and improve with increased data availability.

## 1. Introduction

Efficiently sampling the configurational distribution of high-dimensional molecular systems is a grand challenge in statistical mechanics and computational science (Tuckerman, 2023; Frenkel & Smit, 2023). A key area of interest is the sampling of rare, transition events between two stable configurations, such as in chemical reactions or protein folding (Bolhuis et al., 2002; Dellago et al., 2002). This task, broadly known as transition path sampling (TPS), is challenging due to the presence of energy barriers, which create a substantial difference in timescales between rare events and the fastest dynamical motions of the system (e.g., bond vibrations). This has inspired a rich line of literature on enhanced sampling techniques (Torrie & Valleau, 1977; Swendsen & Wang, 1986; Laio & Parrinello, 2002b; Laio & Gervasio, 2008; Tiwary & Parrinello, 2013; Valsson & Parrinello, 2014). More recently, machine learning (ML) based methods have gained popularity for accelerating TPS by learning a control drift term to bias a system towards a desired target state (Sipka et al., 2023; Holdijk et al., 2024; Du et al., 2024; Seong et al., 2024). However, these approaches rely on highly specialized training procedures and fail to exploit the growing quantity of atomistic simulation and structural data (Bank, 1971; Vander Meersche et al., 2024; Lewis et al., 2024), and the increasing availability of high-quality atomistic conformational generative models (Abramson et al., 2024; Lewis et al., 2024; Jing et al., 2024a; Zheng et al., 2024).

Generative models can produce unbiased, independent samples from atomistic conformational ensembles (Noé et al., 2019; Zheng et al., 2024; Jing et al., 2024a) and have shown the potential to generalize across chemical space (Klein & Noé, 2024; Lewis et al., 2024). However, they have not been directly used for TPS due to the use of uncorrelated states during training. In this work, we propose a conceptually simple post-training method to repurpose generative models to perform TPS in a zero-shot manner. Our core idea exploits the fact that generative models based on denoising diffusion (Ho et al., 2020) and flow matching (Lipman et al., 2022) induce a set of stochastic Langevin dynamics on the data manifold, governed by their learned *score* function. Drawing inspiration from statistical mechanics, the probability of paths sampled from these dynamics can be characterized by a quantity known as the Onsager-Machlup (OM) action functional (Onsager & Machlup, 1953). As a result, we can identify high-probability paths between arbitrary points on the data manifold directly

---

[*]Equal contribution, random order. [1]Dept. of EECS, UC Berkeley [2]Faculty of Mathematics and Physics, Charles University [3]Done prior to joining Amazon [4]Dept. of Chemical and Biomolecular Engineering, UC Berkeley [5]Applied Mathematics and Computational Research, LBNL. Correspondence to: Sanjeev Raja, Aditi S. Krishnapriyan <{sanjeevr, aditik1}@berkeley.edu>.

*Proceedings of the 42$^{nd}$ International Conference on Machine Learning*, Vancouver, Canada. PMLR 267, 2025. Copyright 2025 by the author(s).

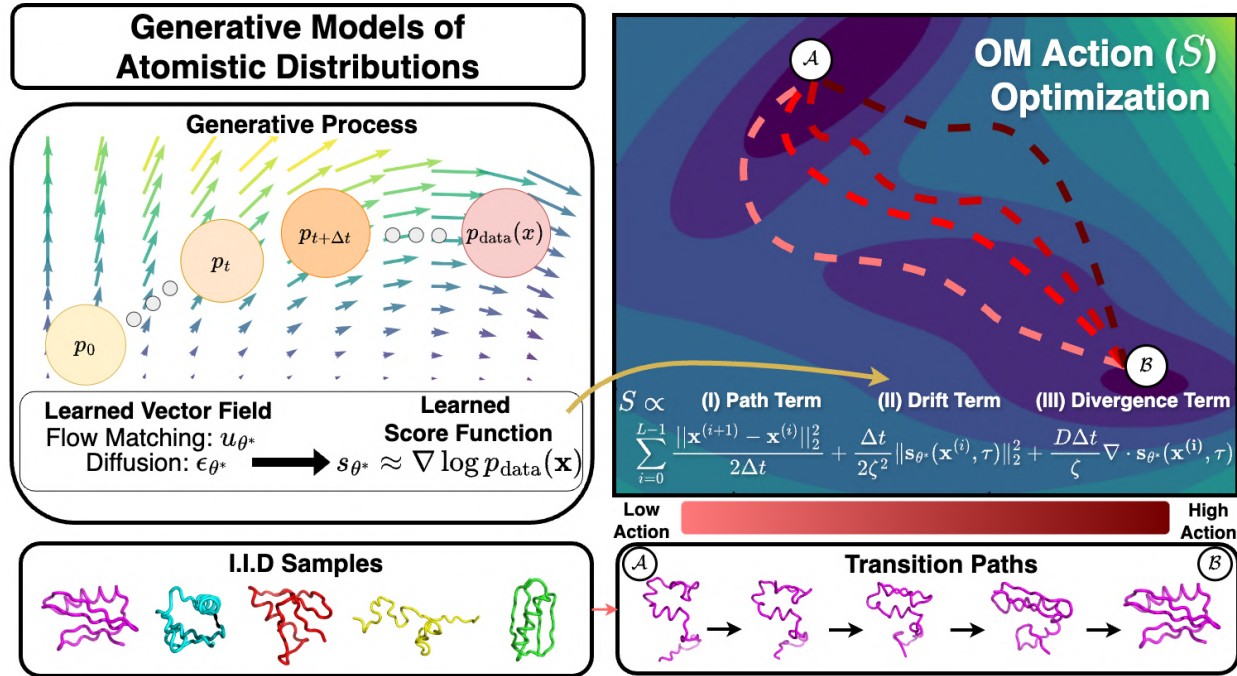

Figure 1. **Proposed Onsager-Machlup Action Optimization Schematic.** **(Left)** Atomistic generative models produce statistically independent samples via integration along a learned vector field, from which a score estimate, $s_{\theta^*} \approx \nabla \log p_{\text{data}}(\mathbf{x})$, can be extracted. The score can be interpreted as a drift term in the stochastic dynamics induced by the generative model. **(Right)** This connection can be leveraged to repurpose atomistic generative models to find high-probability transition paths between samples on the data manifold by minimizing the OM action functional, Eq. (14). The OM action has three terms which prioritize (I) low distance between adjacent points on the discretized path, (II) low-norm drifts, and (III) convexity of the underlying energy landscape. The variables are as follows: path point $i$ ($\mathbf{x}^{(i)}$), trajectory length ($L$), timestep ($\Delta t$), friction ($\zeta$), diffusion ($D$), latent time ($\tau$). The underlying score $s_{\theta^*}$ remains frozen throughout OM optimization. Although valid for any data distribution, in the special case of Boltzmann-distributed data, our approach has the natural interpretation of transition path sampling on a potential energy surface with an atomistic force field.

from first principles by minimizing the OM action with gradient-based optimizers. In the specific case where the data are Boltzmann-distributed, the learned score approximates the underlying atomistic force field (Arts et al., 2023), and our approach has the direct, physical interpretation of TPS on a potential energy surface at a finite temperature.

Our approach has a number of key advantages:

1. **Scalability:** We do not require any training procedure specific to TPS, and instead exploit pre-trained generative models. This makes our approach scalable as models and datasets continue to grow, and generalizable to new systems without retraining.

2. **Flexibility:** We can incorporate physical parameters such as the time horizon and temperature when sampling paths without modifying the underlying generative model.

3. **Efficient Diversity:** We leverage the stochasticity of generative models to produce diverse candidate paths more efficiently than traditional methods.

We validate our approach on systems of increasing complexity. Starting with the 2D Müller-Brown potential (Section

5.1), we build intuition and show accurate estimation of reaction rates and committor functions using our sampled transition paths. On alanine dipeptide (Section 5.2), our method recovers accurate free energy barriers and outperforms metadynamics and shooting algorithms in efficiency. For fast-folding proteins (Section 5.3), OM action minimization with diffusion or flow-matching models yields transition path ensembles closely aligned with reference MD at a fraction of the cost of traditional MD. Finally, we show that OM optimization with generative models trained on tetrapeptide configurations enables zero-shot TPS on new sequences (Section 5.4). Overall, our work demonstrates the promise of repurposing pre-trained generative models as a general-purpose approach for transition path sampling.

## 2. Related Work

**Transition path sampling.** Traditional approaches to TPS like umbrella sampling (Torrie & Valleau, 1977) and metadynamics (Laio & Parrinello, 2002a) employ biasing potentials along representative collective variables (CV). However, defining suitable CVs is challenging near transition states, even with automated approaches (Sultan & Pande, 2018;

Šípka et al., 2023). Meanwhile, shooting techniques (Mullen et al., 2015; Borrero & Dellago, 2016; Jung et al., 2017; Bolhuis & Swenson, 2021), which use a Metropolis-Hastings criterion to sample trajectories, suffer from slow sampling, high rejection rates, and the need for expensive simulations. While ML approaches, including reinforcement learning (Das et al., 2021; Rose et al., 2021; Singh & Limmer, 2023; Seong et al., 2024; Liang et al., 2023), differentiable simulations (Sipka et al., 2023), and $h$-transform learning (Singh & Limmer, 2023; Du et al., 2024), have been used to design CVs or biasing potentials with promising results, they require expensive sampling procedures, must be retrained for each new system of interest, and do not exploit atomistic simulation data. Minimizing the OM action has been used for TPS in low-dimensional systems (Faccioli et al., 2006; Vanden-Eijnden & Heymann, 2008; Autieri et al., 2009; Fujisaki et al., 2010; Lee et al., 2017), but has faced computational challenges, such as a lack of integration with modern auto-differentiation frameworks and the inability to perform gradient-based optimization for larger systems (a Beccara et al., 2012). To our knowledge, our work is the first to propose a gradient-based optimization of the complete OM action, and to connect it to the stochastic dynamics induced by generative models. For a more extensive review of TPS methods, see Appendix A.

**Atomistic generative models.** Generative models can produce unbiased, independent samples from the configurational ensemble of molecular systems, pioneered by Boltzmann generators (Noé et al., 2019) and since further developed for proteins (Arts et al., 2023; Zheng et al., 2024; Jing et al., 2024a; Lewis et al., 2024; Schreiner et al., 2023), small molecules (Huang et al., 2024; Schneuing et al., 2024; Reidenbach & Krishnapriyan, 2024; Igashov et al., 2024), and materials (Zeni et al., 2023; Zheng et al., 2024; Xie et al., 2022). Generative models are typically trained to match the distribution of atomistic configurations from large-scale datasets, including structural databases (Bank, 1971) and long-timescale MD simulations (Lindorff-Larsen et al., 2011; Vander Meersche et al., 2024). Recent works adapt generative models to produce more diverse samples (Corso et al., 2023), perform rare event sampling (Falkner et al., 2023), perform MD simulations using the connection between diffusion models and force fields (Arts et al., 2023), and learn generative models directly over trajectories (Jing et al., 2024b).

**Interpolations in generative models.** Analagous to TPS, interpolation has been used to evaluate the smoothness and continuity of learned data manifolds and to generate realistic transitions between data points using generative models. While linear interpolation in model latent spaces is known to capture some continuity (Kingma & Welling, 2013; Goodfellow et al., 2014), geometric techniques such as geodesic interpolation and optimal transport (Arjovsky et al., 2017; Arvanitidis et al., 2018; Leśniak et al., 2018; Michelis & Becker, 2021; Struski et al., 2023; Psenka et al., 2024) better align

with the intrinsic manifold structure of the data. Our OM optimization approach can be seen as a novel interpolation mechanism which leverages the inductive bias of stochastic dynamics to generate high-probability transition paths.

## 3. Theory

We now introduce the OM action as a way to compute path probabilities under a particular stochastic differential equation (SDE), and describe its application to score-based generative modeling and transition path sampling.

### 3.1. Probability of paths under stochastic dynamics

We introduce the following constant variance SDE which will underpin our proposed framework for TPS:

$$\dot{\mathbf{x}} = \frac{1}{\zeta}\boldsymbol{\Phi}(\mathbf{x})\mathrm{d}t + \sqrt{2D}\mathrm{d}\mathbf{W_t}, \tag{1}$$

where $\boldsymbol{\Phi}(\mathbf{x}):\mathbb{R}^k \to \mathbb{R}^k$ is a drift function, $\mathbf{W}_t$ is a standard Weiner process, and $D,\zeta > 0$ are scalar constants governing diffusion noise levels and damping respectively. We consider drifts which can be written as the gradient of a scalar: $\boldsymbol{\Phi}(\mathbf{x}) = -\frac{\partial\phi(\mathbf{x})}{\partial\mathbf{x}}$, where $\phi(\mathbf{x}) : \mathbb{R}^k \to \mathbb{R}$. By solving a Fokker-Planck equation for the time-varying state distribution $p(\mathbf{x},t)$, we can obtain the probability of a path $\mathbf{x}(\cdot) = \{\mathbf{x}(t)\}_{t=0}^1$ sampled from the SDE in Eq. (1) (see Appendix C for complete details):

$$P(\mathbf{x}(\cdot)) \propto e^{(-S[\mathbf{x}(\cdot)])}. \tag{2}$$

To maximize this probability with respect to a path, we can equivalently minimize the negative log probability $S$, which is called the Onsager-Machlup action functional. This is the stochastic analogue of the well-known principle of least action from optics and quantum mechanics (Rojo et al., 2018). Since we only consider discretized paths in this work, we focus on the discretized form of the OM action:

> **Definition 3.1.** For a discrete path $\mathbf{X} = \{\mathbf{x}^{(0)},...,\mathbf{x}^{(L)}\}$ generated by the SDE in Eq. (1) with drift $\boldsymbol{\Phi}$ and timestep size $\Delta t$, the discretized form of the **Onsager-Machlup action functional** is given by:
>
> $$S[\mathbf{X}] = \frac{1}{2D}\left(A[\mathbf{X}] + B[\mathbf{X}] + C[\mathbf{X}]\right),$$
>
> $$A[\mathbf{X}] = \frac{1}{2\Delta t}\sum_{i=0}^{L-1}\left\|\mathbf{x}^{(i+1)} - \mathbf{x}^{(i)}\right\|_2^2,$$
>
> $$B[\mathbf{X}] = \frac{\Delta t}{2\zeta^2}\sum_{i=1}^{L-1}\left\|\boldsymbol{\Phi}(\mathbf{x}^{(i)})\right\|_2^2, \tag{3}$$
>
> $$C[\mathbf{X}] = \frac{D\Delta t}{\zeta}\sum_{i=1}^{L-1}\nabla\cdot\boldsymbol{\Phi}(\mathbf{x}^{(i)}).$$

The three summands of the OM action each have an intuitive interpretation. Term $A$ encourages smooth transitions along the discretized path. Term $B$ encourages paths remaining in regions with low-norm drifts, which are equilibria or saddle points of the underlying dynamics. Finally, term $C$ encourages paths to remain in regions with low divergence of the drift, which can be interpreted as entropically favoring regions of convexity in the landscape of $\phi$, where dynamics are more stable. The parameters $\zeta, \Delta t$, and $D$ control the relative contribution of these three terms in a physically intuitive manner. For instance, at larger values of $\Delta t$, the contribution of term $A$ is diminished, consistent with the intuition that larger "jumps" are more probable with larger timesteps. In the limiting case of negligible diffusivity $D$ (analogous to temperature, see Appendix C), the divergence term can be omitted, yielding the Truncated OM Action:

> **Definition 3.2.** The **truncated OM action** of a discretized path $\mathbf{X}$ is given by:
>
> $$S[\mathbf{X}] = \frac{1}{2D}\big(A[\mathbf{X}] + B[\mathbf{X}]\big). \qquad (4)$$

In this work, we use both the truncated and full OM actions depending on the particular system considered.

### 3.2. Transition path sampling in molecular systems

The connection between the SDE in Eq. (1) and TPS is straightforward. Formally, in TPS we consider $d$-dimensional molecular systems with $N_p$-many particles interacting under a potential energy function $U(\mathbf{x}): \mathbb{R}^{N_p \times d} \to \mathbb{R}$, where $\mathbf{x} \in \Omega$ is a configuration of the system, and $\Omega \in \mathbb{R}^{N_p \times d}$ is the configuration space. The goal of TPS is to, for the given system and temperature, find most likely paths $\left\{\mathbf{x}^{(i)}\right\}_{0 \leq i \leq L}$ over a time horizon $T_p$, where $\Delta t$ is the simulation timestep and $L = \frac{T_p}{\Delta t}$ is the number of discretization points in the trajectory. The trajectory traverses the two endpoints $\mathbf{x}^{(0)} \in \mathcal{A} \subset \Omega$, $\mathbf{x}^{(L)} \in \mathcal{B} \subset \Omega$, where $\mathcal{A}, \mathcal{B}$ typically represent distinct minima on the potential energy surface $U(\mathbf{x})$. The underlying particle dynamics are governed by the SDE in Eq. (1), known in this context as overdamped Langevin dynamics. The scalar $\phi$ and gradient-based drift $\mathbf{\Phi}$ terms have a clear physical interpretation as the potential energy and forces, respectively:

$$\phi(\mathbf{x}) := U(\mathbf{x}), \qquad (5)$$
$$\mathbf{\Phi}(\mathbf{x}) := \mathbf{F}(\mathbf{x}) = -\nabla U(\mathbf{x}). \qquad (6)$$

If the forces $\mathbf{F}(\mathbf{x})$ are known, the OM action (Eq. (3)) can be used to compute the probability of paths connecting endpoints $\mathbf{x}^{(0)}, \mathbf{x}^{(L)}$ under the governing dynamics.

### 3.3. Score-based generative modeling

We now describe the connection between the SDE in Eq. (1) and generative models, namely denoising diffusion and flow

matching. Specifically, we show that these models induce a set of stochastic dynamics whose drift is given by their learned *score* function. This provides a powerful framework to reason about high-probability transition paths.

#### 3.3.1. STOCHASTIC DYNAMICS UNDER DENOISING DIFFUSION MODELS.

*Denoising diffusion probabilistic models* (DDPM) (Ho et al., 2020) are a class of score-based generative models that learn how to de-noise corrupted samples. The DDPM objective Eq. (25) is closely linked to the score-matching objective (Vincent, 2011; Song & Ermon, 2019) for training a score model $s_\theta(\mathbf{x}, \tau): \mathbb{R}^k \times \mathbb{R}^+ \to \mathbb{R}^k$ parameterized by $\theta$:

$$\mathcal{L}_{\mathrm{SM}}(\theta) = \mathbb{E}_{\tau, \mathbf{x} \sim p_\tau}\left[\|s_\theta(\mathbf{x}, \tau) - \nabla\log(p_\tau(\mathbf{x}))\|_2^2\right], \qquad (7)$$

where $\nabla\log p_\tau(\mathbf{x})$ is the score of the noised marginal distribution $p_\tau$ at time $\tau$. This establishes the connection between the score and the optimal noise model with parameters $\theta^*$: $\epsilon_{\theta^*}(\mathbf{x}, \tau) \propto -\nabla\log p_\tau(\mathbf{x})$. See Appendix B.2 for detailed statements and proofs. In order to use the OM action to compute path probabilities with a DDPM, we must construct a surrogate SDE in the form of Eq. (1), such that paths under this SDE have high likelihood under the data distribution used to train the model. While the denoising (i.e., sampling) process of a DDPM (see Appendix B.1) may appear to be a natural candidate, a closer inspection reveals that it is unsuitable, as it optimizes for different likelihoods at different points of the trajectory. A large portion of the denoising trajectory thus has low likelihood under the data distribution. Therefore, we need to consider an alternative approach.

**Iterative denoising and noising as a candidate SDE.** Another hypothesis for constructing an SDE is to leverage the process of iterative one-step denoising and noising at a fixed time marginal $\tau$ of the diffusion process. Intuitively, this balances the likelihood-maximizing drift of the denoising step with the stochasticity of the noising step. Specifically, we consider the following iterated denoise-noise updates, where $\epsilon_\theta(\mathbf{x}, \tau)$ is the trained denoising model from the DDPM:

$$\mathbf{x}^{(i,\mathrm{mid})} = \frac{1}{\sqrt{1-\beta_\tau}}\left(\mathbf{x}^{(i)} - \frac{\beta_\tau}{\sqrt{1-\bar{\alpha}_\tau}}\epsilon_\theta(\mathbf{x}^{(i)}, \tau)\right) + \sqrt{\beta_\tau}z, \qquad (8)$$

$$\mathbf{x}^{(i+1)} = \sqrt{1-\beta_\tau}\mathbf{x}^{(i,\mathrm{mid})} + \sqrt{\beta_\tau}z', \qquad (9)$$

where $z, z' \sim \mathcal{N}(0, I)$, $\alpha, \beta$ denote the usual diffusion model noise schedule variables, and $\bar{\alpha}_\tau = \prod_{i=1}^\tau \alpha_i$. Combining the two updates yields a single update equivalent in distribution, writing $s_\theta(\mathbf{x}, \tau) = -(1/\sqrt{1-\bar{\alpha}_{\tau-1}})\epsilon_\theta(\mathbf{x}, \tau)$, yields:

$$\mathbf{x}^{(i+1)} = \mathbf{x}^{(i)} + \frac{\beta_\tau\sqrt{1-\bar{\alpha}_{\tau-1}}}{\sqrt{1-\bar{\alpha}_\tau}}s_\theta(\mathbf{x}^{(i)}, \tau) + \sqrt{2\beta_\tau - \beta_\tau^2}z, \qquad (10)$$

where $z \sim \mathcal{N}(0,I)$. Taking the continuum limit of DDPM sampling discretization steps to infinity, we get that $\beta_\tau \to 0$, and noting that $2\beta_\tau - \beta_\tau^2 \approx 2\beta_\tau$ and $\bar{\alpha}_\tau \to \bar{\alpha}_{\tau-1}$ at this limit, we see that Eq. (10) is an Euler-Maruyama discretization, with timestep $\beta_\tau$, of the following SDE:

$$d\mathbf{x} = \mathbf{s}_\theta(\mathbf{x},\tau)\mathrm{d}t + \sqrt{2}\mathrm{d}\mathbf{W_t}. \tag{11}$$

Note that the above construction holds for any $\tau \in \{1,...,T_d\}$ where $T_d$ is the maximum diffusion time. A similar derivation can be found in Arts et al. (2023) for $\tau = 0$.

Eq. (11) is equivalent (up to constants) to Eq. (1):

$$\phi(\mathbf{x}) \propto -\log p_\tau(\mathbf{x}), \tag{12}$$
$$\mathbf{\Phi}(\mathbf{x}) \propto \mathbf{s}_\theta(\mathbf{x},\tau) \approx \nabla \log p_\tau(\mathbf{x}). \tag{13}$$

This means that, as introduced in Eq. (2) and Eq. (3), the likelihood of discrete paths $\mathbf{X} = \left(\mathbf{x}^{(t)}\right)_{0 \le t \le L}$ under the constructed SDE Eq. (11) can be evaluated via the OM action defined by the trained DDPM score $\mathbf{s}_{\theta^*}(\mathbf{x},\tau)$:

$$S(\mathbf{X};\theta^*) = \frac{1}{2D}\left(\sum_{i=0}^{L-1}\frac{1}{2\Delta t}\left\|\mathbf{x}^{(i+1)} - \mathbf{x}^{(i)}\right\|_2^2 \right.$$
$$\left. + \frac{\Delta t}{2\zeta^2}\left\|\mathbf{s}_{\theta^*}(\mathbf{x}^{(i)},\tau)\right\|_2^2 + \frac{D\Delta t}{\zeta}\nabla\cdot\mathbf{s}_{\theta^*}(\mathbf{x}^{(i)},\tau)\right), \tag{14}$$

where $\zeta = 1$, $D = 1$, and $\Delta t = \beta_\tau$. Note that $\beta_\tau$ can be effectively tuned as a hyperparameter by changing the sampling discretization fidelity. In our specific scenario of TPS, we can set these parameters to physically interpretable values, with $\Delta t$, $\zeta$, and $D$ corresponding to the MD simulation timestep, friction coefficient, and diffusion coefficient, respectively (see Section 3.3.3).

### 3.3.2. EXTENSION TO FLOW MATCHING

We now link OM action-minimization with a broader class of generative models beyond DDPM. Flow matching models (Lipman et al., 2022) are a natural choice due to their strong performance in generative modeling tasks across modalities (Jing et al., 2024a; Polyak et al., 2024). Similarly to DDPM, flow matching generates samples through a repeated integration process over a learned vector field. For affine flows considered in this work, the training objective for the learned velocity field $u_\theta(x,\tau)$ takes the form,

$$\mathcal{L}_{\mathrm{FM}}(\theta) = \mathbb{E}_{\tau,\mathbf{x}\sim p_\tau}\left[\|u_\theta(\mathbf{x},\tau) - u_\tau(\mathbf{x})\|_2^2\right], \tag{15}$$

$$u_\tau(\mathbf{x}) = \mathop{\mathbb{E}}_{\mathbf{x}_0\sim p_0,\mathbf{x}_1\sim p_1}[\dot{\alpha}_\tau\mathbf{x}_1 + \dot{\sigma}_\tau\mathbf{x}_0 \,|\, \mathbf{x} = \alpha_\tau\mathbf{x}_1 + \sigma_\tau\mathbf{x}_0], \tag{16}$$

where $p_0$ and $p_1$ are the source and target distributions, $\alpha_\tau, \sigma_\tau : [0,1] \to [0,1]$ define a curve from $\mathbf{x}_0$ to $\mathbf{x}_1$:

$\alpha_0 = \sigma_1 = 0$, $\alpha_1 = \sigma_0 = 1$, and $\alpha_\tau, -\sigma_\tau$ are both strictly increasing functions.

While extracting the learned score $\mathbf{s}_\theta$ from DDPM is straightforward via $\mathbf{s}_\theta(\mathbf{x},\tau) = -(1/\sqrt{1-\bar{\alpha}_{\tau-1}})\epsilon_\theta(\mathbf{x},\tau)$, it is less clear how to do so for flow matching. However, we note the following:

1. By Eq. (7), the targets for the denoising model $\epsilon_\theta(\mathbf{x},\tau)$ in DDPM are equivalently the negative scores of the noised distribution, $-\nabla\log p_\tau(\mathbf{x})$.

2. The targets $u_t(\mathbf{x})$ for the flow matching model $u_\theta(\mathbf{x},\tau)$ can be converted to scores of the flow marginal distribution $\nabla\log p_\tau(\mathbf{x})$ through the following formula (see Appendix B.3 for the proof):

$$\nabla\log p_\tau(\mathbf{x}) = \frac{\alpha_\tau}{\dot{\sigma}_\tau\sigma_\tau\alpha_\tau - \dot{\alpha}_\tau\sigma_\tau^2}\left(\frac{\dot{\alpha}_\tau}{\alpha_\tau}\mathbf{x} - u_\tau(\mathbf{x})\right). \tag{17}$$

We can thus extract an approximate score $\mathbf{s}_{\theta^*}^{\mathrm{FM}}(\mathbf{x},\tau)$ from a trained flow matching model $u_{\theta^*}(\mathbf{x},\tau)$ via,

$$\mathbf{s}_\theta^{\mathrm{FM}}(\mathbf{x},\tau) = \frac{\alpha_\tau}{\dot{\sigma}_\tau\sigma_\tau\alpha_\tau - \dot{\alpha}_\tau\sigma_\tau^2}\left(\frac{\dot{\alpha}_\tau}{\alpha_\tau}\mathbf{x} - u_{\theta^*}(\mathbf{x},\tau)\right). \tag{18}$$

By inserting $\mathbf{s}_{\theta^*}^{\mathrm{FM}}(\mathbf{x},\tau)$ into the denoise-noise process defined in Eq. (11), we again obtain an SDE of the form of Eq. (1). Hence, we can use the OM action to compute the log-probabilities of paths between arbitrary datapoints.

### 3.3.3. PHYSICAL INTERPRETATION FOR BOLTZMANN-DISTRIBUTED DATA

By assuming underlying dynamics of the form Eq. (11), our framework combining generative models and the OM action can be used to compute the log-probabilities of paths between samples from *any* data distribution with a well-defined score. However, in the special case where the data used to train the generative model are Boltzmann distributed, i.e., $p(\mathbf{x}) \propto \exp\left(-\frac{U(\mathbf{x})}{k_B T}\right)$, where $k_B$ is Boltzmann's constant and $T$ is the temperature, the learned score $\mathbf{s}_{\theta^*}(\mathbf{x},\tau=0) : \mathbb{R}^{N_p \times d} \to \mathbb{R}^{N_p \times d}$ is interpretable as a physical, atomistic force field:

$$\mathbf{s}_{\theta^*}(\mathbf{x},\tau=0) \approx \nabla\log p(\mathbf{x}) \propto -\nabla U(\mathbf{x}) = \mathbf{F}(\mathbf{x}). \tag{19}$$

The constructed dynamics in Eq. (11) at $\tau = 0$ reduce to overdamped Langevin dynamics governing particles on a potential energy surface. Thus, we can directly set the constants in the OM action ($\Delta t$, $\zeta$, $D$, $T$) to physical values used in MD simulations for a given atomistic system. Under our OM framework, finding high-probability paths between points on a Boltzmann-distributed data manifold thus directly aligns with the conventional notion of TPS for atomistic systems.

# 4. Repurposing Atomistic Generative Models via Action Minimization

We now introduce our OM optimization approach to produce high probability transition paths between atomistic structures using pre-trained generative models. Given a pre-trained generative model with a fixed score function $\mathbf{s}_{\theta^*}$, and two atomistic configurations, $\mathbf{x}^{(0)}, \mathbf{x}^{(L)} \in \mathbb{R}^{N_p \times d}$ where $\mathbf{x}^{(0)} \in \mathcal{A}, \mathbf{x}^{(L)} \in \mathcal{B}$, we aim to sample a transition path $\mathbf{X} = \left\{ \mathbf{x}^{(i)} \right\}_{i \in [0,L]}$ consisting of $L$ discrete steps. Our core inductive bias is to interpret candidate paths as realizations of the denoise-noise SDE in Eq. (11), enabling tractable computation and optimization of path log-likelihoods via the OM action. Our approach proceeds in three primary steps: 1) computing an initial guess path, 2) performing OM optimization, and (in some cases) 3) decoding the optimized path back to the configurational space $\Omega$. Algorithm 1 summarizes the procedure without the decoding step.

**Computing an initial guess path.** The choice of initial path connecting the endpoints $\mathbf{x}^{(0)} \in \mathcal{A}$ and $\mathbf{x}^{(L)} \in \mathcal{B}$ is crucial in determining the quality of the subsequently optimized path. Naïve linear interpolations in configurational space are likely unphysical, since plausible atomistic structures typically lie on a highly non-convex, low-dimensional manifold of $\Omega$. Instead, we opt to linearly interpolate at a *latent* level $\tau_{\text{initial}}$ of our pre-trained generative model, which is known to produce samples closer to the data manifold (Ho et al., 2020). After interpolation, we can either (1) decode the interpolated path back into configurational space and subsequently optimize the OM action there, or (2) directly optimize the OM action at the latent level $\tau_{\text{initial}}$. Both are justified, since the proposed dynamics in Eq. (11) are valid for any $\tau$. See Appendix G for complete details.

**Optimization of the OM action.** Starting from the initial guess, we find paths which have high probability (low action) under the SDE in Eq. (11) induced by the generative model. The optimization problem simply becomes,

$$\mathbf{X}^* = \underset{\mathbf{X} = \left\{ \mathbf{x}^{(i)} \right\}_{i \in [0,L]}}{\text{argmin}} \; S[\mathbf{X}; \theta^*], \qquad (20)$$

where $S[\cdot; \theta^*]$ is the generative model action in Eq. (14) (computed with the frozen, pre-trained score $\mathbf{s}_{\theta^*}$), and $\mathbf{x}^{(0)}$ and $\mathbf{x}^{(L)}$ are kept fixed. We approximate this minimum via gradient descent on the path until the action is converged. Since $S[\cdot; \theta^*]$ is a discretized integral, the entire trajectory is optimized in parallel, which is amenable to multi-device acceleration. In line with previous work (Arts et al., 2023), we find that a small, nonzero value of $\tau_{\text{opt}}$ leads to better alignment with the true forces, and thus treat it as a hyper-parameter (see Appendix G). To accelerate computation of the divergence term in the OM action, we use the Hutchinson estimator (Hutchinson, 1989)(see Appendix G for details).

**Algorithm 1** Onsager-Machlup Transition Path Optimization with Generative Models

1: **Input:**
2: Optimization time $\tau_{\text{opt}}$
3: Generative model with time-conditional score $\mathbf{s}_{\theta^*}(\cdot, \tau_{\text{opt}})$
4: Two atomistic configurations $\mathbf{x}^{(0)} \in \mathcal{A} \subset \mathbb{R}^{N_p \times d}$, $\mathbf{x}^{(L)} \in \mathcal{B} \subset \mathbb{R}^{N_p \times d}$
5: **Compute** initial guess $\mathbf{X} = \left\{ \mathbf{x}^{(0)}, ..., \mathbf{x}^{(L)} \right\} = $ InitialGuess$(\mathbf{x}^{(0)}, \mathbf{x}^{(L)}, \tau_{\text{initial}})$ (Algorithm 2)
6: **while** not converged **do**
7:     **Compute** the OM action, $S[\mathbf{X}; \theta^*]$, with the learned vector field $\mathbf{s}_{\theta^*}(\cdot, \tau_{\text{opt}})$, using Eq. (14).
8:     **Update** $\mathbf{X} \leftarrow$ optimizer$(\mathbf{X}, \nabla_{\mathbf{X}} S[\mathbf{X}; \theta^*])$ (keeping the endpoints $\mathbf{x}^{(0)}, \mathbf{x}^{(L)}$ fixed)
9: **end while**

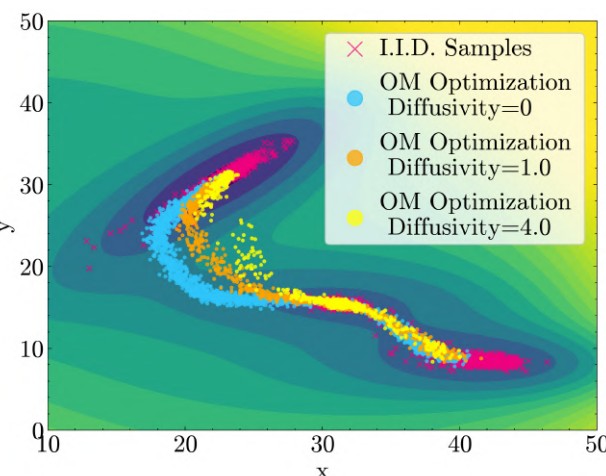

*Figure 2.* **OM optimization with a diffusion model on the 2D Müller-Brown potential.** Individual points along the OM-optimized paths are shown as dots. Increasing the diffusivity $D$ causes the path to cross at a higher barrier. An equivalent number of I.I.D samples (red) fails to sample the transition region.

**Decoding back to configurational space.** If OM-optimization was performed at a non-zero latent time $\tau_{\text{initial}}$, we decode the final path, obtained after $K$ iterations of gradient descent, back to the configurational space. If optimization was performed in configuration space (i.e., decoding was already done in the first step), then this step is skipped.

# 5. Results

We now present the results of our OM optimization approach for TPS with pre-trained generative models. In all cases other than the Müller-Brown potential (Section 5.1), we perform OM optimization in configuration space, so we skip the final decoding step and use physical parameter values. In Appendix F, we demonstrate a use case beyond generative modeling, namely using a classical force field to generate

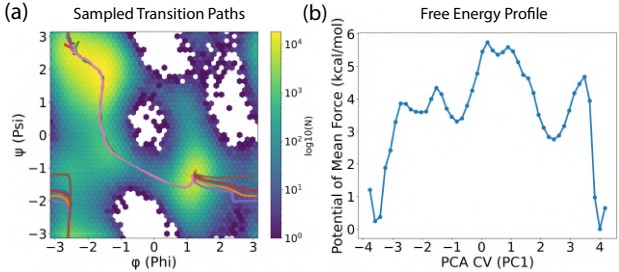

*Figure 3.* **OM optimization and free energy barrier estimation on alanine dipeptide using a pre-trained diffusion model. (a)** Sampled alanine dipeptide transition paths from OM optimization with a pretrained diffusion model, overlaid on a Ramachandran plot of sample density at 600K. **(b)** Free energy barrier for a selected path, estimated via umbrella sampling, is approximately 6 kcal/mol, in line with our metadynamics calculations (I).

*Table 1.* **Benchmarking the speed of OM optimization against traditional enhanced sampling baselines on alanine dipeptide.** We report the number of force field evaluations required per 1,000 step transition path (for OM optimization we instead report the number of score function evaluations). OM optimization requires significantly fewer evaluations and thus lower computational cost per path than the traditional approaches.

| Method | CVs | # FF Evals / Path ($\downarrow$) | Runtime / Path ($\downarrow$) |
|---|---|---|---|
| MCMC (Two-Way Shooting) | No | $\geq$ 1B | $\geq$ 100 hours |
| Metadynamics | Yes | 1M | 10 hours |
| **OM Opt. (Diffusion Model) (ours)** | No | **10K** | **50 min** |

all-atom transition paths for Chignolin and Trp-Cage.

### 5.1. 2D Müller-Brown potential

We first demonstrate our method on the 2D Müller-Brown (MB) potential (Müller & Brown, 1979), a canonical test system for TPS with a global minimum and two local minima, separated by saddle points.

**Problem setup.** Using a denoising diffusion model pretrained on samples from the underlying potential energy surface, we generate transitions between the two deepest energy minima using OM optimization. We generate the initial guess path via linear interpolation at $\tau_{\text{initial}} = 8$ (the maximum diffusion model time is $T_d = 1,000$). We perform 200 steps of OM optimization directly at $\tau_{\text{initial}} = 8$ using $\tau_{\text{opt}} = 8$, and finally decode the path back to the data distribution via the denoising process.

**Results.** As shown in Fig. 2, OM optimization with the truncated action yields a transition path (shown in blue) between the energy minima which passes through the lowest energy barrier. Due to the stochastic decoding process, the samples around the transition path exhibit natural diversity. Increasing the diffusion coefficient $D = \frac{k_B T}{\gamma M}$ results in qualitatively different transition paths (shown in orange and yellow). Samples are more concentrated in the three

energy minima, and the path crosses a higher energy barrier, consistent with the larger scale of thermal fluctuations at increased temperatures. Meanwhile, 2500 i.i.d. samples (shown in red) from the diffusion model sample only the three energy minima, and fail to sample the transition region. This provides a proof-of-concept that our OM optimization procedure can be used to repurpose generative models to sample transition paths without specialized training. See Appendix H for additional results and analysis on the MB potential, including with a flow matching model.

**Committor and Transition Rate Estimation.** Committor functions and rates are fundamental quantities in the study of transition events (Vanden-Eijnden et al., 2006). The committor function $q(\mathbf{x})$ is defined as the probability that a trajectory initiated at $\mathbf{x}^{(0)} = \mathbf{x}$ reaches the target state $\mathcal{B}$ before the starting state $\mathcal{A}$. Transition paths obtained via OM optimization can be used as an enhanced sampling method to accurately compute the committor function, and subsequently the transition rates, on the Müller-Brown potential (see Appendix H for complete details). Specifically, we initiate MD simulations from points along the OM-optimized paths shown in Fig. 2 to collect a dataset of samples $\mathcal{D}_{\text{train}}$. We then train a committor function $q_\theta(\mathbf{x})$ by solving a functional optimization problem given by the Backward Kolmogorov Equation (BKE) over the sampled points, and finally compute an estimate of the reaction rate using the trained model via the relation (Vanden-Eijnden et al., 2006):

$$k_\theta = \frac{k_B T}{\gamma} \langle |\nabla_\mathbf{x} q_\theta(\mathbf{x})|^2 \rangle_\Omega \approx \hat{\mathbb{E}}_{\mathbf{x} \sim \mathcal{D}_{\text{train}}} |\nabla_\mathbf{x} q_\theta(\mathbf{x})|^2, \quad (21)$$

where $\langle \cdot \rangle_\Omega$ denotes an ensemble average over the configurational space $\Omega$ and $\hat{\mathbb{E}}$ denotes an empirical mean . Using this procedure, we obtain a transition rate estimate of $1.3 \times 10^{-5}$, compared with the true rate of $5.4 \times 10^{-5}$. It is often challenging to compute the reaction rate even within the correct order of magnitude (Rotskoff et al., 2022; Hasyim et al., 2022), suggesting the promise of our OM optimization method to enable accurate rate estimation on more challenging systems.

### 5.2. Alanine dipeptide

Alanine dipeptide is a classic benchmark system for TPS, with 22 atoms and CVs described by the dihedral angles $\phi, \psi$.

**Problem setup.** We start with a denoising diffusion model pre-trained on samples from the alanine dipeptide potential energy surface. Using this model, we perform OM optimization to find transition paths between the two standard minima defined in the CV space.

**Results.** We successfully sample two likely transition paths between the metastable basins (Fig. 3a). In Table 1, we benchmark the computational efficiency of OM optimization against traditional enhanced sampling techniques, namely metadynamics (Laio & Gervasio, 2008) and MCMC-based

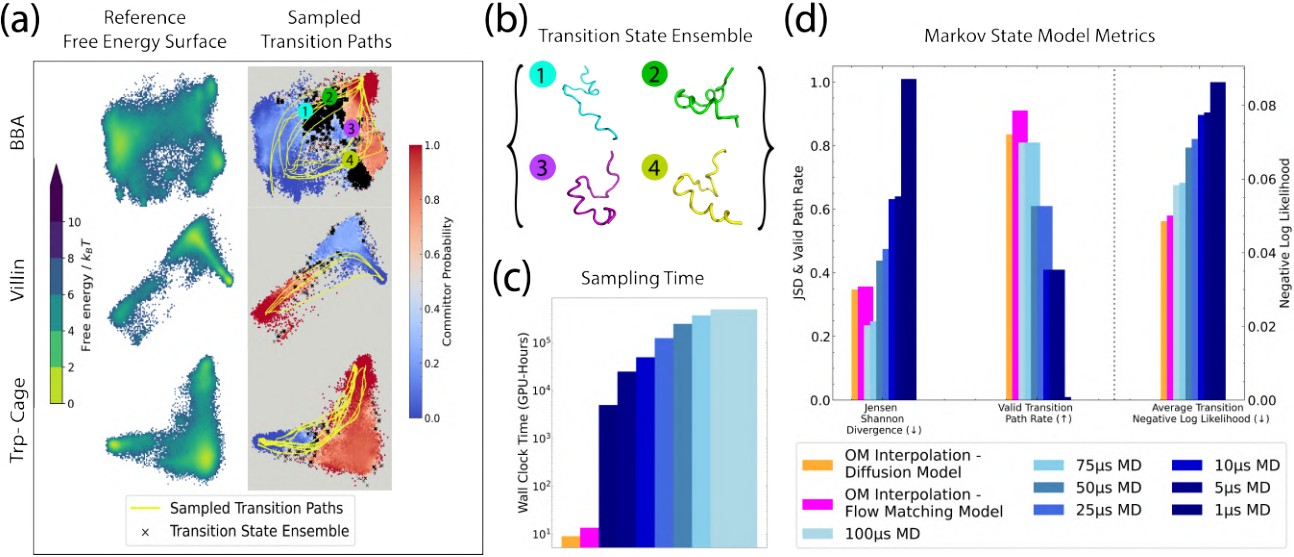

*Figure 4.* **OM optimization with diffusion and flow matching models trained on coarse-grained, fast-folding proteins. (a)** Reference free energy surfaces of fast-folding proteins, alongside transition paths produced by OM optimization (yellow) overlaid against the landscape of the empirically computed committor function $q(\mathbf{x})$. The transition state ensemble (black) is the set $\{\mathbf{x} : 0.45 \leq q(\mathbf{x}) \leq 0.55\}$. **(b)** Samples from the predicted transition state ensemble of BBA. **(c)** Runtime of OM optimization and varying lengths of unbiased MD simulation of BBA. MD simulation is performed with the diffusion model score as an approximate force field, as in Arts et al. (2023). **(d)** MSM results averaged across 5 fast-folding proteins. Comparisons are with respect to reference, unbiased MD simulation of varying lengths. Plotted are the Jensen-Shannon Divergence (a measure of distributional dissimilarity) between the sampled and reference path MSM state distributions, fraction of sampled paths which are valid (i.e., have non-zero probability under the reference MSM), and average transition negative log likelihood under the reference MSM (indicating the realism of the paths).

two-way shooting (Bolhuis & Swenson, 2021), and find that OM optimization is considerably more efficient. As shown in Fig. 3b, we can also use the transition paths resulting from OM optimization as a natural CV for umbrella sampling (Torrie & Valleau, 1977), from which we can accurately and efficiently estimate free energy profiles along the transition path. See Appendix I for complete details on model training, OM optimization, and free energy calculations.

### 5.3. Fast-folding coarse-grained proteins

We next consider proteins exhibiting fast dynamical transitions, for which millisecond-scale, reference MD simulations were performed in Lindorff-Larsen et al. (2011).

**Problem setup.** We adopt a coarse-graining (CG) scheme which represents each amino acid with the position of its $C_\alpha$ atom. We utilize the pre-trained diffusion models from Arts et al. (2023), and we train our own flow matching models. Separate models are trained for each protein. To facilitate analysis and interpretation of results, we divide the conformational space into discrete states and make use of Markov State Models (MSMs) (Prinz et al., 2011; Noé et al., 2013) to obtain state transition probabilities. Similar to Jing et al. (2024b), we evaluate the quality of transition paths by discretizing them over the MSM states and computing the following metrics (see Appendix J for complete details):

1. **Transition negative log likelihood.** The negative log likelihood of MSM state transitions under the reference MSM, averaged over all paths with non-zero probability.

2. **Fraction of valid paths.** The fraction of paths with non-zero probability under the reference MSM.

3. **Jensen-Shannon divergence.** The JSD (distributional dissimilarity) between the distribution of states visited by the generated paths and those sampled from the reference MSM.

We compute these metrics for our generated transition paths, as well as for $1-100\mu s$ subsets of the reference MD simulations [1]. To compare wall-clock time for generating transition paths, since reference simulations were performed at all-atom resolution and their speeds are unavailable, we follow Arts et al. (2023) and run coarse-grained MD using the generative model's learned score function as an approximate force field.

**Results.** As shown in Fig. 4a, OM optimization yields diverse transition paths which intuitively pass through high density regions of the free energy landscape, projected onto the two slowest Time Independent Component (TIC)

---

[1] We did not consider enhanced sampling baselines for the fast-folding proteins due to their reliance on an energy function, which is generally not available for our level of coarse-graining.

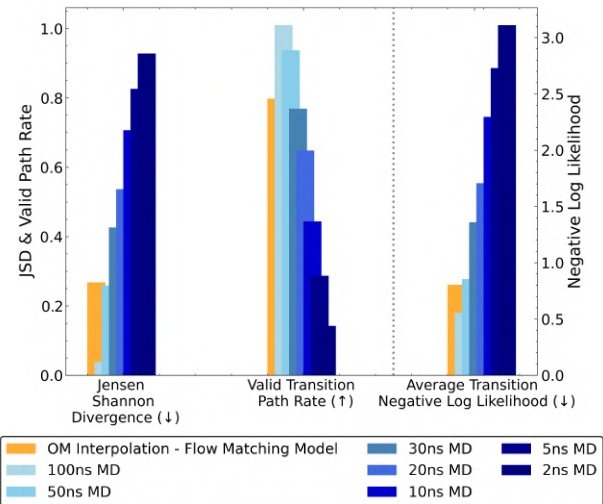

*Figure 5.* **OM optimization on unseen tetrapeptide sequences.** OM optimization with a flow matching model yields transition paths which compare strongly with variable-length MD simulations, indicating the generalization potential of our approach. Plotted are the Jensen Shannon Divergence (a measure of distributional dissimilarity) between state distributions visited by the reference and generated paths, the fraction of valid (non-zero probability) paths under the reference MSM, and the negative log likelihood of state transitions under the reference MSM (indicating path realism).

(Noé et al., 2013) axes. We can also vary the physical parameters of the OM action (e.g., the time horizon $T_p$) to obtain paths traversing different regions of phase space (see Appendix J). For the BBA protein, the paths robustly sample the transition state ensemble, empirically defined by the level set $\{\mathbf{x} : 0.45 \leq q(\mathbf{x}) \leq 0.55\}$ of the committor function (Fig. 4b). Sampling transition paths of the BBA protein with OM optimization requires considerably less wall-clock time than using the diffusion model's learned score as a coarse-grained force field and performing unbiased MD simulations (Fig. 4c). Across all proteins and both classes of generative model (diffusion and flow matching), OM optimization yields a higher percentage of valid paths and lower transition negative log likelihood under the reference MSM, compared with unbiased, reference MD simulations of any of the considered lengths up to 100 $\mu$s (Fig. 4d). The JSD is also lower than any MD simulation length up to 50 $\mu$s, indicating that the sampled paths traverse a similar distribution of MSM states as the reference simulations. See Appendix J for comparisons to transition trajectories found in the reference, unbiased MD simulations.

**Robustness to sparse data in transition regions.** To simulate the scenario in which transition states are not well-represented in the training data, we retrain diffusion models on datasets from which 99% of configurations with committor probability between 0.1 and 0.9 are removed. OM optimization is still able to sample plausible transition

paths using this data-starved model (see Appendix J for more details), suggesting that our approach can be useful even if the underlying data distribution is not an exhaustively sampled Boltzmann distribution.

### 5.4. Generalization to new tetrapeptides

As a final evaluation, we consider all-atom tetrapeptide systems, which exhibit interesting dynamics and pose the challenge of generalization to held-out amino acid sequences.

**Problem setup.** We train a flow matching generative model on approximately 3,000 tetrapeptides simulated in Jing et al. (2024b), and apply our OM optimization procedure to generate an ensemble of 16 transition paths for each of 100 tetrapeptides *not seen* during training. We use the same MSM-based metrics as in Section 5.3 to evaluate the quality of generated transition paths.

**Results.** As shown in Fig. 5, OM optimization achieves MSM metrics which are competitive with MD simulations of 50-100 ns, which are considerably more computationally expensive to generate. This suggests the promise of OM optimization to generate transition paths on atomistic systems not explicitly seen during training. See Appendix K for further details and path visualizations.

## 6. Conclusion

We have presented a method to repurpose atomistic generative models for transition path sampling by finding paths over the data manifold which minimize the Onsager-Machlup action under the model's learned score function. Our approach, which requires no TPS-specific training procedure, aligns well with the growing trend of leveraging large-scale, well-tested, general-purpose generative models—a direction already standard in the language and vision communities.

**Limitations.** Our approach does not provably sample the full posterior distribution over paths, as in traditional shooting methods and recent ML approaches (Du et al., 2024). However, we sample diverse paths by exploiting the stochastic generative model encoding and decoding process. Initiating traditional MD or umbrella sampling simulations is another way to explore the potential energy surface around the OM-optimized paths (see Appendix I).

**Future work.** Incorporating OM optimization into larger generative models trained on more diverse data (Lewis et al., 2024; Jing et al., 2024a) is a natural area for future development. Given the success of large-scale, co-evolutionary modeling of proteins (Jumper et al., 2021), it would be interesting to investigate the extent to which pre-training generative models on large structural databases enables TPS on unseen systems. More broadly, OM action-minimization could be a powerful framework to generate interpolation paths in a variety of data modalities, including images, videos, and audio.

## Impact Statement

This paper presents work whose goal is to advance the field of Machine Learning. There are many potential societal consequences of our work, none which we feel must be specifically highlighted here.

## Acknowledgments

The authors thank Rasmus Lindrup, Aditya Singh, Muhammad Hasyim, Kranthi Mandadapu, Simon Olsson, Soojung Yang, Lukáš Grajciar, Johannes Dietschreit and David Limmer for helpful discussions that benefited this paper, as well as Bowen Jing for assistance with reproducing the min-flux endpoints for the tetrapeptide evaluations. The authors acknowledge support of this work from the U.S. Department of Energy, Office of Science, Energy Earthshot initiatives as part of the Center for Ionomer-based Water Electrolysis at Lawrence Berkeley National Laboratory under Award Number DE-AC02-05CH11231. This work was also supported by the Toyota Research Institute as part of the Synthesis Advanced Research Challenge. MP and MŠ were supported by the Czech Science Foundation, project 23-05736S.

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

## A. Extended Related Work: Transition Path Sampling

A rich landscape of tools has been developed for TPS, for which we refer to existing surveys for a more exhaustive description (Bolhuis et al., 2002; Dellago et al., 2002; Vanden-Eijnden et al., 2010). Traditional shooting methods perturb the initial or intermediate states of a known trajectory to generate new trajectories via a Metropolis-Hasting criterion (Mullen et al., 2015; Borrero & Dellago, 2016; Jung et al., 2017; Bolhuis & Swenson, 2021). These often suffer from high rejection rates, correlated samples, and the need for expensive molecular dynamics (MD) simulations during sampling. Another class of methods is based on adding an adjustable biasing potential to enhance the sampling of slow events, which includes umbrella sampling (Torrie & Valleau, 1977), metadynamics (Laio & Parrinello, 2002a), and more advanced techniques such as eABF (Darve & Pohorille, 2001). These approaches require a carefully constructed, low-dimensional mapping of the problem via collective variables (CVs), which can be challenging, particularly when the characterization of the system around the transition state is uncertain. Attempts were made to design CVs with ML methods (Sultan & Pande, 2018; Rogal et al., 2019; Chen & Ferguson, 2018; Sun et al., 2022; Šípka et al., 2023), yet they remain a challenge for many-atom systems. ML approaches have also been used to learn the biasing potential directly, including approaches based on stochastic optimal control (Holdijk et al., 2024; Yan et al., 2022), differentiable simulation (Sipka et al., 2023), reinforcement learning (Das et al., 2021; Rose et al., 2021; Singh & Limmer, 2023; Seong et al., 2024; Liang et al., 2023), and $h$-transform learning (Singh & Limmer, 2023; Du et al., 2024). In all of these approaches, unlimited access to the underlying potential energy and force field are assumed, but samples from the underlying data distribution are not available. As a result, the methods must be retrained from scratch for every new system of interest. Additionally, expensive, simulation-based training procedures are often employed, limited scalability to larger systems. Interpolation-based methods, such as the Nudged Elastic Band (NEB) method (Henkelman et al., 2000) and the spring method (Dellago et al., 1998), introduce springs between images to construct transition paths (spring method) or to directly locate saddle points (NEB). However, a significant challenge for both approaches lies in generating an initial guess, which is inherently unknown a priori. Among interpolation-like approaches, the Onsager-Machlup (OM) action has also been explored for TPS. Due to the limited availability of automatic differentiation techniques at the time, Laplacian operators were consistently avoided. This restriction limited its application to very low-dimensional problems (Vanden-Eijnden & Heymann, 2008; Fujisaki et al., 2010), or led to the development of Laplace-free action formulations (Lee et al., 2017).

## B. Proofs for Score-Related Generative Model Objectives

For the sake of readability, we replicate proofs showing that both the training objectives for DDPM and flow matching models are equivalent to training against the score of a noised version of the data distribution (or in the case of flow matching, an invertible polynomial transformation of this score). See Vincent (2011) and Lipman et al. (2024) for example proofs for DDPM and flow matching respectively.

### B.1. A note on the DDPM reverse process

The sampling process for a DDPM can be written as the terminal condition $\mathbf{x}^{(0)}$ of the following process:

$$\mathbf{x}^{(T_d)} \sim \mathcal{N}(0, I), \tag{22}$$

$$\mathbf{x}^{(i-1)} = \frac{1}{\sqrt{1-\beta_\tau}}\left(\mathbf{x}^{(i)} + \frac{\beta_\tau}{\sqrt{1-\bar{\alpha}_\tau}}\mathbf{s}_\theta(\mathbf{x}^{(i)}, i)\right) + \sqrt{\beta_\tau} z. \tag{23}$$

While this process is definable as an Euler-Maruyama discretization of an SDE, it is not well suited for optimization over trajectories over the data distribution, since the vector field $\mathbf{s}_\theta(\mathbf{x}^{(i)}, i)$ is changing throughout the trajectory, and iterates near the noise $i = T_d$ will not necessarily follow dynamics determined by the data distribution.

### B.2. DDPM and score matching

Below is a proof for the equivalence of standard DDPM training to score matching.

**Theorem B.1** (DDPM-Score Matching Equivalence). *Let $p_{data}(x_0)$ be the data distribution, and let $x_\tau$ be the noised variable defined through the forward process:*

$$x_\tau = \sqrt{\bar{\alpha}_\tau}x_0 + \sqrt{1-\bar{\alpha}_\tau}\epsilon, \quad \tau \sim \text{Unif}(\{1, ..., T_d\}), \quad x_0 \sim p_{data}, \quad \epsilon \sim \mathcal{N}(0, I), \tag{24}$$

*where $\bar{\alpha}_\tau \in (0, 1)$. Let $p_\tau(x_\tau) = \int p_{data}(x_0)\mathcal{N}(x_\tau; \sqrt{\bar{\alpha}_\tau}x_0, (1-\bar{\alpha}_\tau)I)dx_0$ be the marginal distribution of $x_\tau$. Then the DDPM objective, defined as the following:*

$$\mathcal{L}_{DDPM}(\theta) \overset{\text{def}}{=} \mathbb{E}_{\tau, x_0, \epsilon}\left[\|\epsilon - \epsilon_\theta(x_\tau, \tau)\|_2^2\right], \tag{25}$$

*satisfies the following equality:*

$$\mathcal{L}_{DDPM}(\theta) = \mathbb{E}_{\tau,x_\tau}\left[(1-\bar{\alpha}_\tau)\|\nabla_{x_\tau}\log p_\tau(x_\tau) - s_\theta(x_\tau,\tau)\|_2^2\right] + C, \tag{26}$$

*where $s_\theta(x_\tau,\tau) \overset{\text{def}}{=} -\epsilon_\theta(x_\tau,\tau)/\sqrt{1-\bar{\alpha}_\tau}$, and $C$ is a constant independent of $\theta$.*

*Proof.* We begin with the DDPM training objective:

$$\mathcal{L}_{\text{DDPM}}(\theta) = \mathbb{E}_{\tau,x_0,\epsilon}\left[\|\epsilon - \epsilon_\theta(x_\tau,\tau)\|_2^2\right]. \tag{27}$$

The core of the desired result is Tweedie's formula, which relates Gaussian-based denoising to the score of the noised distribution. For any random variable $z$ generated as $z = \mu + \sigma\eta$ where $\mu$ is an arbitrary random vector, $\eta \sim \mathcal{N}(0,I)$, and $\sigma > 0$, Tweedie's formula gives the following posterior expectation:

$$\mathbb{E}[\mu|z] = z + \sigma^2\nabla_z\log p(z), \tag{28}$$

where $p(z) = \int \mathcal{N}(z;\mu,\sigma^2 I)p(\mu)d\mu$ is the full marginal distribution of $z$. In the forward process, $x_\tau$ is generated via $x_\tau = \sqrt{\bar{\alpha}_\tau}x_0 + \sqrt{1-\bar{\alpha}_\tau}\epsilon$, which corresponds to:

$$\mu = \sqrt{\bar{\alpha}_\tau}x_0, \quad \sigma = \sqrt{1-\bar{\alpha}_\tau}, \quad z = x_\tau, \quad \eta = \epsilon. \tag{29}$$

Here, $\mu$ is a random variable (dependent on $x_0$), not a fixed parameter. Applying Tweedie's formula to the marginal distribution $p_\tau(x_\tau)$, we obtain:

$$\mathbb{E}\left[\sqrt{\bar{\alpha}_\tau}x_0\big|x_\tau\right] = x_\tau + (1-\bar{\alpha}_\tau)\nabla_{x_\tau}\log p_\tau(x_\tau). \tag{30}$$

Dividing through by $\sqrt{\bar{\alpha}_\tau}$ gets the posterior of the original sample $x_0$:

$$\mathbb{E}[x_0|x_\tau] = \frac{x_\tau}{\sqrt{\bar{\alpha}_\tau}} + \frac{1-\bar{\alpha}_\tau}{\sqrt{\bar{\alpha}_\tau}}\nabla_{x_\tau}\log p_\tau(x_\tau). \tag{31}$$

From the forward process definition, we rewrite in terms of $\epsilon$:

$$\epsilon = \frac{x_\tau - \sqrt{\bar{\alpha}_\tau}x_0}{\sqrt{1-\bar{\alpha}_\tau}}, \tag{32}$$

and take conditional expectations given $x_\tau$ to get the following:

$$\mathbb{E}[\epsilon|x_\tau] = \frac{x_\tau - \sqrt{\bar{\alpha}_\tau}\mathbb{E}[x_0|x_\tau]}{\sqrt{1-\bar{\alpha}_\tau}}, \tag{33}$$

$$= \frac{x_\tau - \sqrt{\bar{\alpha}_\tau}\left(\frac{x_\tau}{\sqrt{\bar{\alpha}_\tau}} + \frac{1-\bar{\alpha}_\tau}{\sqrt{\bar{\alpha}_\tau}}\nabla_{x_\tau}\log p_\tau(x_\tau)\right)}{\sqrt{1-\bar{\alpha}_\tau}}, \tag{34}$$

$$= \frac{x_\tau - x_\tau - (1-\bar{\alpha}_\tau)\nabla_{x_\tau}\log p_\tau(x_\tau)}{\sqrt{1-\bar{\alpha}_\tau}}, \tag{35}$$

$$= -\sqrt{1-\bar{\alpha}_\tau}\nabla_{x_\tau}\log p_\tau(x_\tau). \tag{36}$$

Using the law of total expectation, we can expand the DDPM loss conditioned on $x_\tau,\tau$:

$$\mathcal{L}_{\text{DDPM}}(\theta) = \mathbb{E}_{\tau,x_\tau}\left[\mathbb{E}_\epsilon\left[\|\epsilon - \epsilon_\theta(x_\tau,\tau)\|_2^2\,|\,x_\tau,\tau\right]\right]. \tag{37}$$

For any random vector $\xi$, $\mathbb{E}[\|\xi - c\|^2]$ is minimized when $c = \mathbb{E}[\xi]$. We can then make the following bias-variance decomposition:

$$\mathbb{E}_\epsilon\left[\|\epsilon - \epsilon_\theta(x_\tau,\tau)\|_2^2\,|\,x_\tau,\tau\right] = \|\mathbb{E}[\epsilon\,|\,x_\tau,\tau] - \epsilon_\theta(x_\tau,\tau)\|_2^2 + \mathbb{E}\left[\|\epsilon - \mathbb{E}[\epsilon\,|\,x_\tau,\tau]\|_2^2\,|\,x_\tau,\tau\right]. \tag{38}$$

Since the variance term is independent of $\theta$, substituting $\mathbb{E}[\epsilon\,|\,x_\tau,\tau]$ and factoring out $-\sqrt{1-\bar{\alpha}_\tau}$ leads to:

$$\mathcal{L}_{\text{DDPM}}(\theta) = \mathbb{E}_{\tau,x_\tau}\left[(1-\bar{\alpha}_\tau)\left\|\nabla_{x_\tau}\log p_\tau(x_\tau) - \left(-\frac{\epsilon_\theta(x_\tau,\tau)}{\sqrt{1-\bar{\alpha}_\tau}}\right)\right\|_2^2\right] + C. \tag{39}$$

By defining $s_\theta(x_\tau,\tau) := -\epsilon_\theta(x_\tau,\tau)/\sqrt{1-\bar{\alpha}_\tau}$, we obtain the score matching objective. $\qquad\square$

## B.3. Flow matching and score matching

We now provide proof for the flow matching setting, showing that the training objective is also similar to a score matching objective, with a simple transformation between the flow matching targets and the scores.

**Theorem B.2** (Flow Matching – Score Matching Conversion). *Let $p_{data}(x_0)$ be the data distribution, and let $x_\tau$ be the noised variable defined through the interpolation process:*

$$x_\tau = \alpha_\tau x_1 + \sigma_\tau x_0, \quad \tau \sim \text{Unif}([0,1]), \quad x_1 \sim p_{data}, \quad x_0 \sim \mathcal{N}(0,I), \tag{40}$$

*where $\alpha_\tau, \sigma_\tau : [0,1] \to [0,1]$ are strictly increasing and decreasing functions respectively that satisfy $\alpha_0 = \sigma_1 = 0$, $\alpha_1 = \sigma_0 = 1$. Let $p_\tau(x_\tau) = \int p_{data}(x_0)\mathcal{N}(x_\tau; \alpha_\tau x_1, \sigma_\tau^2 I)dx_0$ be the marginal distribution of $x_\tau$. Then the flow matching objective, defined as the following:*

$$\mathcal{L}_{FM}(\theta) \overset{\text{def}}{=} \mathbb{E}_{\tau,x_0,x_1}\left[\|u_\theta(x_\tau,\tau) - v_\tau(x_0,x_1)\|_2^2\right], \tag{41}$$

*where $v_\tau(x_0,x_1) = \dot{\alpha}_\tau x_1 + \dot{\sigma}_\tau x_0$ the instance-wise curve velocity. The flow matching objective then satisfies the following equalities:*

1. *We can equivalently train against targets of the unconditional velocities $u_\tau(\mathbf{x}) = \mathbb{E}_{\mathbf{x}_0 \sim p_0, \mathbf{x}_1 \sim p_1}[\dot{\alpha}_t \mathbf{x}_1 + \dot{\sigma}_t \mathbf{x}_0 \mid \mathbf{x} = \alpha_t \mathbf{x}_1 + \sigma_t \mathbf{x}_0]$:*

   $$\mathcal{L}_{FM}(\theta) = \mathbb{E}_{\tau,x_0,x_1}\left[\|u_\theta(x_\tau,\tau) - u_\tau(x_\tau)\|_2^2\right] + C, \tag{42}$$

   *where $C$ is some constant independent of $\theta$, and*

2. *The equality $\nabla_x \log p_\tau(x) = \frac{\dot{\alpha}_\tau}{\alpha_\tau}x - \frac{\dot{\sigma}_\tau \sigma_\tau \alpha_\tau - \dot{\alpha}_\tau \sigma_\tau^2}{\alpha_\tau}u_\tau(x)$ holds, allowing us to write the flow matching objective in terms of the score:*

   $$\mathcal{L}_{FM}(\theta) = \mathbb{E}_{\tau,x_0,x_1}\left[\left\|u_\theta(x_\tau,\tau) - \left(\frac{\dot{\alpha}_\tau}{\alpha_\tau}x_\tau - \frac{\dot{\sigma}_\tau \sigma_\tau \alpha_\tau - \dot{\alpha}_\tau \sigma_\tau^2}{\alpha_\tau}\nabla_{x_\tau}\log p_\tau(x_\tau)\right)\right\|_2^2\right] + C. \tag{43}$$

*Proof.* For part 1, we can expand the flow matching loss integrand by telescoping with respect to $u_\tau(x_\tau)$:

$$\|u_\theta(x_\tau,\tau) - v_\tau(x_0,x_1)\|^2 = \|u_\theta(x_\tau,\tau) - u_\tau(x_\tau) + u_\tau(x_\tau) - v_\tau(x_0,x_1)\|^2, \tag{44}$$

$$= \|u_\theta(x_\tau,\tau) - u_\tau(x_\tau)\|^2 + 2\langle u_\theta(x_\tau,\tau) - u_\tau(x_\tau), u_\tau(x_\tau) - v_\tau(x_0,x_1)\rangle + \|u_\tau(x_\tau) - v_\tau(x_0,x_1)\|^2. \tag{45}$$

Since $\mathbb{E}_{\tau,x_0,x_1}\|u_\tau(x_\tau) - v_\tau(x_0,x_1)\|^2$ is constant with respect to $\theta$, it suffices to show the following:

$$\mathbb{E}_{\tau,x_0,x_1}\langle u_\theta(x_\tau,\tau) - u_\tau(x_\tau), u_\tau(x_\tau) - v_\tau(x_0,x_1)\rangle = 0. \tag{46}$$

Note that by definition we have the following relation between $u$ and $v$:

$$\mathbb{E}_{x_0,x_1}[v_\tau(x_0,x_1) \mid x_\tau,\tau] = \mathbb{E}_{x_0,x_1}[\dot{\alpha}_\tau x_1 + \dot{\sigma}_\tau x_0 \mid x_\tau,\tau] = u_\tau(x_\tau). \tag{47}$$

We can then write the following by expanding the expectation using the tower rule:

$$\mathbb{E}_{\tau,x_0,x_1}\langle u_\theta(x_\tau,\tau) - u_\tau(x_\tau), u_\tau(x_\tau) - v_\tau(x_0,x_1)\rangle = \mathbb{E}_{x_\tau,\tau}\left[\mathbb{E}_{x_0,x_1}[\langle u_\theta(x_\tau,\tau) - u_\tau(x_\tau), u_\tau(x_\tau) - v_\tau(x_0,x_1)\rangle \mid x_\tau,\tau]\right], \tag{48}$$

$$= \mathbb{E}_{x_\tau,\tau}[\langle u_\theta(x_\tau,\tau) - u_\tau(x_\tau), u_\tau(x_\tau) - \mathbb{E}_{x_0,x_1}[v_\tau(x_0,x_1) \mid x_\tau,\tau]\rangle], \tag{49}$$

$$= \mathbb{E}_{x_\tau,\tau}[\langle u_\theta(x_\tau,\tau) - u_\tau(x_\tau), \mathbf{0}\rangle], \tag{50}$$

$$= 0. \tag{51}$$

This concludes part 1. Note that the interpolations $x_\tau = \alpha_\tau x_1 + \sigma_\tau x_0$ also follow proper form for Tweedie's formula, allowing us to write the following:

$$\mathbb{E}[\alpha_\tau x_1 \mid x_\tau,\tau] = x_\tau + \sigma_\tau^2 \nabla_x \log p_\tau(x_\tau), \tag{52}$$

$$\mathbb{E}[x_1 \mid x_\tau,\tau] = \frac{1}{\alpha_\tau}x_\tau + \frac{\sigma_\tau^2}{\alpha_\tau}\nabla_x \log p_\tau(x_\tau). \tag{53}$$

Noting that $x_0 = \frac{x_\tau - \alpha_\tau x_1}{\sigma_\tau}$, we can write $u_\tau$ as the following:

$$u_\tau(x) = \mathop{\mathbb{E}}_{x_0, x_1} \left[ \dot\alpha_t x_1 + \dot\sigma_t x_0 \,|\, x_\tau = x\tau \right], \tag{54}$$

$$= \dot\alpha_t \mathop{\mathbb{E}}_{x_0, x_1} \left[ x_1 \,|\, x_\tau = x, \tau \right] + \dot\sigma_t \mathop{\mathbb{E}}_{x_0, x_1} \left[ x_0 \,|\, x_\tau = x, \tau \right], \tag{55}$$

$$= \frac{\dot\sigma_\tau}{\sigma_\tau} x + \left( \dot\alpha_\tau - \frac{\alpha_\tau \dot\sigma_\tau}{\sigma_\tau} \right) \mathop{\mathbb{E}}_{x_0, x_1} \left[ x_1 \,|\, x_\tau = x, \tau \right], \tag{56}$$

$$= \frac{\dot\sigma_\tau}{\sigma_\tau} x + \left( \dot\alpha_\tau - \frac{\alpha_\tau \dot\sigma_\tau}{\sigma_\tau} \right) \left( \frac{1}{\alpha_\tau} x + \frac{\sigma_\tau^2}{\alpha_\tau} \nabla_x \log p_\tau(x) \right), \tag{57}$$

$$= \frac{\dot\alpha_\tau}{\alpha_\tau} x - \frac{\dot\sigma_\tau \sigma_\tau \alpha_\tau - \dot\alpha_\tau \sigma_\tau^2}{\alpha_\tau} \nabla_x \log p_\tau(x). \tag{58}$$

This proves the desired relation between $u_\tau$ and $\nabla \log p_\tau$, and plugging into part 1 achieves the desired flow matching loss equality.

$\square$

## C. Derivation of the Onsager-Machlup action

### C.1. Overdamped Langevin dynamics

We start the description of our system by formulating the well-known Hamilton equations. The variables we are solving are $\mathbf{x}_i(t) : \mathbb{R}^+ \to \mathbb{R}^d$ and the corresponding momenta $\mathbf{p}_i(t) : \mathbb{R}^+ \to \mathbb{R}^d$ with a constant vector $m_i$ representing the mass of every particle $i$ in the system. Hamiltonian equations are formulated as follows

$$\dot{\mathbf{x}}_i(t) = \frac{\mathbf{p}_i(t)}{m_i},$$
$$\dot{\mathbf{p}}_i(t) = -\frac{\partial U(\mathbf{x}(t))}{\partial \mathbf{x}_i}. \tag{59}$$

While these equations maintain energy and contain no representation of temperature, a modified SDE, with the term $\mathbf{W}(t)$ representing a Wiener process and a damping constant $\gamma$

$$\dot{\mathbf{x}}_i(t) = \frac{\mathbf{p}_i(t)}{m_i},$$
$$\dot{\mathbf{p}}_i(t) = -\frac{\partial U(\mathbf{x}(t))}{\partial \mathbf{x}_i} - \gamma \frac{\mathbf{p}_i(t)}{m_i} + \sqrt{2\gamma k_B T} \mathbf{W}(t), \tag{60}$$

or equivalently in one second order equation:

$$m_i \ddot{\mathbf{x}}_i(t) = -\frac{\partial U(\mathbf{x}(t))}{\partial \mathbf{x}_i} - \gamma m_i \dot{\mathbf{x}}_i + \sqrt{2 m_i \gamma k_B T} \mathbf{W}(t), \tag{61}$$

can now represent a system that experiences thermal fluctuation. Although the original Hamiltonian system is trapped in an energy well forever, the one guided by Langevin dynamics may overcome barriers between wells in finite time.

A question then arises. Of all the possible paths of fixed physical parameters and time that connect two minima, which is the most probable? How do we calculate probabilities and penalize high energy regions or paths that are making too large steps? The answer is provided by Onsager and Machlup in their works (Onsager & Machlup, 1953; Machlup & Onsager, 1953). The second reference handles the full equation Eq. (60), while the first is a reduction to a so-called overdamped state where the term $\ddot{\mathbf{x}}(t)$ can be neglected. After introduction of two auxiliary vector quantities $\zeta_i = m_i \gamma$ and $D_i = \frac{k_b T}{\zeta_i}$ we get the form

$$\dot{\mathbf{x}}_i = -\frac{1}{\zeta_i} \frac{\partial U(\mathbf{x}(t))}{\partial \mathbf{x}_i} + \sqrt{2D_i} \mathbf{W}(t). \tag{62}$$

or equally just with $\mathbf{F}(\mathbf{x}(t)) = -\frac{\partial U(\mathbf{x}(t))}{\partial \mathbf{x}_i}$

$$\dot{\mathbf{x}}_i = \frac{1}{\zeta_i} \mathbf{F}(\mathbf{x}(t)) + \sqrt{2D_i} \mathbf{W}(t). \tag{63}$$

Further, we will follow a more general setting of the Langevin equation consistent with Eq. (1). To recall:

$$\dot{\mathbf{x}} = \frac{1}{\zeta}\mathbf{\Phi}(\mathbf{x})dt + \sqrt{2D}d\mathbf{W}, \tag{64}$$

and

$$\phi(\mathbf{x}) := U(\mathbf{x}) \tag{65}$$
$$\mathbf{\Phi}(\mathbf{x}) := \mathbf{F}(\mathbf{x}) = -\nabla U(\mathbf{x}) \tag{66}$$

### C.2. Most probable path under Langevin dynamics

Considering a single particle (for more particle systems see e.g. (Kappler & Adhikari, 2020)), since Eq. (1) is a stochastic differential equation, we can also write a partial differential equation for the probability density of the particle guided by these equations. In this case it is a well-known Fokker-Planck equation (note $\frac{\partial}{\partial \mathbf{x}}$ of a vector will be understood as a divergence operator to save space)

$$\frac{\partial P}{\partial t} = -\frac{\partial\left(\frac{\mathbf{\Phi}}{\zeta}P\right)}{\partial \mathbf{x}} + \frac{\partial}{\partial \mathbf{x}}\left(D\frac{\partial P}{\partial \mathbf{x}}\right). \tag{67}$$

As we will consider only potential forces in this work, let us denote $\mathbf{\Phi}(\mathbf{x}) = -\frac{\partial\phi(\mathbf{x})}{\partial \mathbf{x}}$. Now we will split the derivation into two parts.

**1. $\mathbf{\Phi} = 0$:**

Let us consider the solution in the following form:

$$P(\mathbf{x},t\,|\,\mathbf{x}_0) = (4\pi Dt)^{-\frac{3}{2}}e^{\frac{-(\mathbf{x}-\mathbf{x}_0)^2}{4Dt}}. \tag{68}$$

Then for the sequence of points in space and time $(\mathbf{x}^1,t^1),(\mathbf{x}^2,t^2),...(\mathbf{x}^N,t^N)$ we can write the following probability:

$$P(\mathbf{x}^1,t^1\,|\,\mathbf{x}^2,t^2\,|\cdots|\,\mathbf{x}^N,t^N) = \prod_{j=1}^{N}P(\mathbf{x}^j,t^j-t^{j-1}\,|\,\mathbf{x}_0). \tag{69}$$

Let us denote $t^j - t^{j-1} = \epsilon$ as we pass through a continuum limit in time. The probability can be rewritten by plugging in a solution Eq. (68) into

$$\prod_{j=1}^{N}P(\mathbf{x}^j,t^j-t^{j-1}\,|\,\mathbf{x}_0) = (4\pi D\epsilon)^{-\frac{3}{2}N}\exp\left(-\frac{1}{4D\epsilon}\sum_{j=1}^{N}(\mathbf{x}_j-\mathbf{x}_{j-1})^2\right). \tag{70}$$

To make sure we can pass into the limit let us rewrite

$$\epsilon^{-\frac{3}{2}} = e^{-\frac{3}{2}\ln\epsilon}. \tag{71}$$

We now focus on the argument of the $\exp$ function. We can modify it to the form

$$\frac{1}{4D}\sum_{j=1}^{N}\left(\frac{\mathbf{x}_j-\mathbf{x}_{j-1}}{\epsilon}\right)^2\epsilon. \tag{72}$$

By passing into the limit $N\to\infty$ and realizing that epsilon can be rewritten by its definition to $\epsilon = \frac{t}{N}$, we get the following integral form:

$$\frac{1}{4D}\int_0^t(\dot{\mathbf{x}})^2dt. \tag{73}$$

Note however, using the identity $a^b = e^{b\ln a}$, the first part of the product goes to infinity:

$$\lim_{N\to\infty}\left(4\pi D\frac{t}{N}\right)^{-\frac{3}{2}N} = \infty, \tag{74}$$

evaluation of this limit directly would be too hasty. One must consider the probability derived in the broader context of integration across the path. In that case, the constant will serve to normalize the probability. The fact that it does not depend on $\mathbf{x}$ also means that the probability of the path does not, relative to other paths, depend on this prefactor, and only the exponential part is important. To find out more about precise mathematical justifications, we refer the reader to (Gel'fand & Yaglom, 1960). We shall denote the constant before exponential as $C$ from now on because, as it is not dependent on $\mathbf{x}$, it will not influence our calculations. The final probability of a path is then given as follows:

$$P(\mathbf{x},t) = C \exp\left(-\frac{1}{4D}\int_0^t (\dot{\mathbf{x}}(s))^2 ds\right). \tag{75}$$

To maximize the likelihood of the path we clearly need to minimize the action

$$S_0(\mathbf{x}(t)) = \frac{1}{4D}\int_0^t (\dot{\mathbf{x}}(s))^2 ds. \tag{76}$$

Intuitively, the most probable path under no drift is the one that does not move from it's origin. The longer the trajectory, the less probable it is.

**2. $\Phi \neq 0$:**

We will recall the assumption $\Phi = -\nabla\phi(\mathbf{x})$ and shall use a transformation

$$P(\mathbf{x},t\,|\,\mathbf{x}_0) = G(\mathbf{x},t,|\,\mathbf{x}_0)\exp\left(\frac{1}{2D\zeta}\int_{\mathbf{x}(0)}^{\mathbf{x}(t)} \Phi(\mathbf{s})d\mathbf{s}\right), \tag{77}$$

where from the properties of a potential function we can evaluate the integral to

$$-\overline{\phi}(\mathbf{x}) \overset{\text{def}}{=} \frac{1}{2D\zeta}\int_{\mathbf{x}(0)}^{\mathbf{x}(t)} \Phi(\mathbf{s})d\mathbf{s} = \frac{1}{2D\zeta}\int_{\mathbf{x}(0)}^{\mathbf{x}(t)} \Phi(\mathbf{s})d\mathbf{s} = \frac{1}{2D\zeta}(-\phi(\mathbf{x})+\phi(\mathbf{x}_0)). \tag{78}$$

So for clarity:

$$P(\mathbf{x},t) = G(\mathbf{x},t)e^{-\overline{\phi}(\mathbf{x})}, \tag{79}$$

$$G(\mathbf{x},t) = P(\mathbf{x},t)e^{\overline{\phi}(\mathbf{x})}, \tag{80}$$

$$\nabla\overline{\phi}(\mathbf{x}) = \frac{1}{2D\zeta}\nabla\phi(\mathbf{x}). \tag{81}$$

We then plug this transformed function into Eq. (67). We will now derive the equation that $G(\mathbf{x},t)$ has to fulfill. Let us evaluate left-hand side of the Eq. (67)

$$\frac{\partial P(\mathbf{x},t)}{\partial t} = \frac{\partial G(\mathbf{x},t)}{\partial t}e^{-\overline{\phi}(\mathbf{x})}. \tag{82}$$

For the right-hand side lets evaluate first the term:

$$\begin{aligned}
\frac{\partial\left(-\frac{1}{\zeta}\frac{\partial\phi(\mathbf{x})}{\partial\mathbf{x}}P(\mathbf{x},t)\right)}{\partial\mathbf{x}} &= -P(\mathbf{x},t)\frac{1}{\zeta}\frac{\partial^2\phi(\mathbf{x})}{\partial\mathbf{x}^2} - \frac{1}{\zeta}\frac{\partial P(\mathbf{x},t)}{\partial\mathbf{x}}\frac{\partial\phi(\mathbf{x})}{\partial\mathbf{x}}, \\
&= -2DP(\mathbf{x},t)\frac{\partial^2\overline{\phi}}{\partial\mathbf{x}^2} - 2D\frac{\partial P(\mathbf{x},t)}{\partial\mathbf{x}}\frac{\partial\overline{\phi}}{\partial\mathbf{x}}, \\
&= -2DGe^{-\overline{\phi}(\mathbf{x})}\frac{\partial^2\overline{\phi}}{\partial\mathbf{x}^2} - 2D\frac{\partial G}{\partial\mathbf{x}}e^{-\overline{\phi}(\mathbf{x})}\frac{\partial\overline{\phi}}{\partial\mathbf{x}} + 2DP\left(\frac{\partial\overline{\phi}}{\partial\mathbf{x}}\right)^2.
\end{aligned} \tag{83}$$

While the other term can be written as follows:

$$\begin{aligned}
\frac{\partial P(\mathbf{x},t)}{\partial\mathbf{x}} &= \frac{\partial G(\mathbf{x},t)}{\partial\mathbf{x}}e^{-\overline{\phi}} - G(\mathbf{x},t)e^{-\overline{\phi}}\frac{\partial\overline{\phi}}{\partial\mathbf{x}}, \\
&= \frac{\partial G(\mathbf{x},t)}{\partial\mathbf{x}}e^{-\overline{\phi}} - P(\mathbf{x},t)\frac{\partial\overline{\phi}}{\partial\mathbf{x}}.
\end{aligned} \tag{84}$$

The second derivative then with function arguments omitted for brevity, yet remaining the same

$$D\frac{\partial^2 P}{\partial \mathbf{x}^2} = D\frac{\partial^2 G}{\partial \mathbf{x}^2}e^{-\overline{\phi}} - D\frac{\partial G}{\partial \mathbf{x}}e^{-\overline{\phi}}\frac{\partial \overline{\phi}}{\partial \mathbf{x}} - D\frac{\partial P}{\partial \mathbf{x}}\frac{\partial \overline{\phi}}{\partial \mathbf{x}} - DP\frac{\partial^2 \overline{\phi}}{\partial \mathbf{x}^2},$$
$$= D\frac{\partial^2 G}{\partial \mathbf{x}^2}e^{-\overline{\phi}} - 2D\frac{\partial G}{\partial \mathbf{x}}e^{-\overline{\phi}}\frac{\partial \overline{\phi}}{\partial \mathbf{x}} + DP\left(\frac{\partial \overline{\phi}}{\partial \mathbf{x}}\right)^2 - DGe^{-\overline{\phi}}\frac{\partial^2 \overline{\phi}}{\partial \mathbf{x}^2}. \tag{85}$$

After subtracting the terms on the right-hand side, we get the following:

$$\frac{\partial G}{\partial t}e^{-\overline{\phi}} = D\frac{\partial^2 G}{\partial \mathbf{x}^2}e^{-\overline{\phi}} - DGe^{-\overline{\phi}}\left(\frac{\partial \overline{\phi}}{\partial \mathbf{x}}\right)^2 + DGe^{-\overline{\phi}}\frac{\partial^2 \overline{\phi}}{\partial \mathbf{x}^2}. \tag{86}$$

Or written nicely after exponential cancels and we return to $\phi(\mathbf{x})$ from $\overline{\phi}$

$$\frac{\partial G(\mathbf{x},t)}{\partial t} = D\frac{\partial^2 G(\mathbf{x},t)}{\partial \mathbf{x}^2} - G(\mathbf{x},t)\left(\frac{1}{4D}\left(\frac{1}{\zeta}\frac{\partial \phi(\mathbf{x})}{\partial \mathbf{x}}\right)^2 - \frac{1}{2\zeta}\frac{\partial^2 \phi(\mathbf{x})}{\partial \mathbf{x}^2}\right). \tag{87}$$

This is a well studied diffusion-reaction equation

$$\frac{\partial u(\mathbf{x},t)}{\partial t} = D\frac{\partial^2 u(\mathbf{x},t)}{\partial \mathbf{x}^2} - ku(\mathbf{x},t). \tag{88}$$

Notice for $\phi(\mathbf{x}) \equiv 0$ we already solved this equation as it is identical to Fokker-Planck where $F = 0$. Let us call this solution $u_0$. Another observation is that for this equation we can formulate a solution in the form

$$u(\mathbf{x},t) = u_0(\mathbf{x},t)e^{-\int_0^t k(s)ds},$$
$$u_0(\mathbf{x},t) = Ce^{\frac{-(\mathbf{x}-\mathbf{x}_0)^2}{4Dt}}, \tag{89}$$

where $C$ is some arbitrary normalization constant as in the previous solution:

$$G(\mathbf{x},t) = C\exp\left(\frac{-(\mathbf{x}-\mathbf{x}_0)^2}{4D\epsilon} - \int_{\mathbf{x}(0)}^{\mathbf{x}(t)} k(\mathbf{s})d\mathbf{s}\right), \tag{90}$$

and the original $P(\mathbf{x},t)$ using Eq. (77):

$$P(\mathbf{x},t) = C\exp\left(\frac{-(\mathbf{x}-\mathbf{x}_0)^2}{4Dt} + \int_{\mathbf{x}(0)}^{\mathbf{x}(t)}\left(-k(\mathbf{x}(\mathbf{s})) + \frac{1}{2D\zeta}\mathbf{\Phi}(\mathbf{s})\right)d\mathbf{s}\right). \tag{91}$$

Now we repeat the same multiplication of probabilities for small time increments. However, this time, the situation is easier as integrals would simply extend in the sum. Therefore the only limit would be in the first term exactly as done before. The final probability of the path is then as follows:

$$P(\mathbf{x},t) = C\exp\left[-\frac{1}{4D}\int_0^t (\dot{\mathbf{x}})^2 + \left(\frac{1}{\zeta}\frac{\partial \phi}{\partial \mathbf{x}}\right)^2 - \frac{2D}{\zeta}\frac{\partial^2 \phi}{\partial \mathbf{x}^2} - 2\mathbf{\Phi}ds\right]. \tag{92}$$

The negative argument of the exponential will again be an action to minimize:

$$S(\mathbf{x}(t)) = \frac{1}{4D}\int_0^t (\dot{\mathbf{x}})^2 + \left(\frac{1}{\zeta}\frac{\partial \phi}{\partial \mathbf{x}}\right)^2 - \frac{2D}{\zeta}\frac{\partial^2 \phi}{\partial \mathbf{x}^2} - 2\mathbf{\Phi}(\mathbf{s})ds. \tag{93}$$

This can be further modified, by integrating forces along the path and using forces instead of a potential, to the more common form:

$$S(\mathbf{x}(t)) = \frac{1}{2D}(\phi(\mathbf{x}) - \phi(\mathbf{x}_0)) + \frac{1}{4D}\int_0^t \dot{\mathbf{x}}^2 + \left(\frac{1}{\zeta}\frac{\partial \phi}{\partial \mathbf{x}}\right)^2 - \frac{2D}{\zeta}\frac{\partial^2 \phi}{\partial \mathbf{x}^2}ds. \tag{94}$$

This procedure to derive the Onsager-Machlup action is similar to that in (Mauri, 2012).

## D. Fixed endpoints

For the entirety of this paper, we operate with fixed endpoints. This means the actual minimized action will be reduced simply to the following:

$$S(x(t_e)) = \frac{1}{4D} \int_0^t \dot{\mathbf{x}}^2 + \left( \frac{1}{\zeta} \frac{\partial \phi}{\partial \mathbf{x}} \right)^2 - \frac{2D}{\zeta} \frac{\partial^2 \phi}{\partial \mathbf{x}^2} d\mathbf{s}. \tag{95}$$

The first and simplest strategy to keep endpoint constant is to include a penalty in the form

$$L_p = C_{spring} \left[ (\mathbf{x}(0) - \overline{\mathbf{x}}_0)^2 + (\mathbf{x}(t) - \overline{\mathbf{x}}_T)^2 \right]. \tag{96}$$

Interestingly, as verified experimentally, the penalty term effectively works the same as using a more simple and straightforward method. The method of choice was to set the endpoint gradients to 0 manually. Only a couple of points will be affected, and the majority of the trajectory is the same for both approaches.

## E. Numerical discretization

Another aspect to consider is the numerical discretization of the action. (Adib, 2008) discusses the different numerical evaluations of the action stemming from the stochastic nature of the Langevin equation. Namely, the Onsager-Machlup action depends on the discretization convention used for the SDE (Itô or Stratonovich). We consider the Stratonovich convention for this paper, which yields the following Onsager-Machlup action:

$$S(\mathbf{x}_0, \mathbf{x}_1 ... \mathbf{x}_n) = \frac{1}{4D} \sum_{j=1}^{N} - \frac{(\mathbf{x}_j - \mathbf{x}_{j-1})^2}{\Delta t} + \Delta t \left( \frac{\mathbf{\Phi}(\mathbf{x}_j)}{\zeta} \right)^2 + \frac{2D \Delta t}{\zeta} \nabla \cdot \mathbf{\Phi}. \tag{97}$$

If the Itô convention was used, there would not be a Jacobian term $\nabla \cdot \mathbf{\Phi}$ (Cugliandolo & Lecomte, 2017).

To make the multidimensional, multiparticle system discretization clear, we extend the sum along spatial dimensions (index $j$) and we also sum particles (index $k$). The coefficient $\zeta$ is now a vector since it is originally $\zeta_k = \gamma/M_k$ where $M_k$ is the vector of masses. The total action is then,

$$S[\mathbf{x}^{(0)}, ..., \mathbf{x}^{(L)}] = \sum_{i=1}^{L-1} \sum_{j=1}^{N_p} \frac{1}{4D \Delta t} \left\| \mathbf{x}_j^{(i+1)} - \mathbf{x}_j^{(i)} \right\|^2 + \frac{\Delta t}{4D \zeta_j^2} \left\| \mathbf{\Phi}_j(\mathbf{x}^{(i)}) \right\|^2 - \frac{\Delta t}{2 \zeta_j} \nabla \cdot \mathbf{\Phi}_j(\mathbf{x}^{(i)}). \tag{98}$$

## F. Classical Force Fields on All-Atom Proteins

We demonstrate that our OM action optimization framework is broadly useful for transition path sampling even beyond the setting of generative modeling. Specifically, we aim to find all-atom transition paths between the unfolded and folded states of the protein Chignolin and Trp-Cage, using a differentiable PyTorch implementation (Doerr et al., 2020; Sipka et al., 2023) of the Amber *ff14SB*(Maier et al., 2015) forcefield and the *TIP3P* implicit water model. We choose the physical parameters of the OM action to be consistent with commonly used values in molecular simulations (see Table 2). Since we do not have a generative model from which to obtain an initial path guess via latent interpolation as described in 4, we instead employ a **hierarchical unwrapping** warm-up procedure described in the Appendix G to obtain initial paths. As the classical force field is dominated by quadratic terms whose Laplacian is constant and thus uninformative for optimization, we use a zero-temperature approximation and optimize with the Truncated OM action. Using the Truncated action, we obtain a physical transition path of length 2.6ps (shown in Fig. 6). This is much lower than previously reported transition path lengths for Chignolin (Sobieraj & Setny, 2022; Lindorff-Larsen et al., 2011), which can be explained by the fact that our trajectories proceed between the target states without fluctuations that would occur in unbiased simulations. The entire optimization took on the order of hours on one NVIDIA RTX A6000 GPU, including the generation of initial trajectory.

## G. Additional Details on Onsager-Machlup Action Minimization Method

**Initial Path Guess Methods.** We provide a complete description and algorithmic formulation of the initial path guess method using a generative model, mentioned in Section 4.

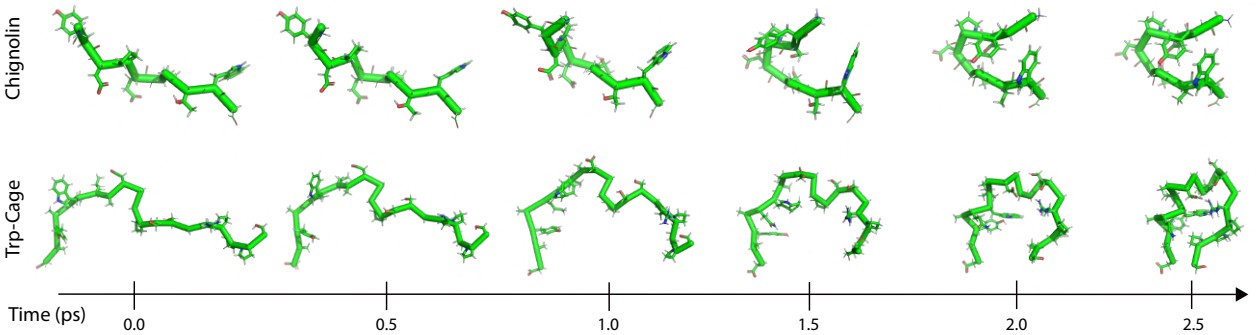

*Figure 6.* Transition paths from OM optimization of all-atom chignolin and trp-cage with a classical force field.

*Table 2.* Hyperparameters used for OM action optimization on all atom proteins

| Hyperparameter | Chignolin - warmup | Chignolin | TRP Cage - warmup | TRP Cage |
|---|---|---|---|---|
| Number of points per path | 40 - 2600 | 2600 | 40 - 2600 | 2600 |
| Action Type | Truncated | Truncated | Truncated | Truncated |
| Optimizer | Adam | Adam | Adam | Adam |
| Learning Rate | $10^{-4}$ | $10^{-5}$ | $10^{-4}$ | $10^{-5}$ |
| Action Timestep ($\Delta t$) | 1 fs | 1 fs | 1 fs | 1 fs |
| Action Friction ($\gamma$) | $10\,\mathrm{ps}^{-1}$ | $10\,\mathrm{ps}^{-1}$ | $10\,\mathrm{ps}^{-1}$ | $10\,\mathrm{ps}^{-1}$ |

Formally, consider a generative model with a non-parametric, probabilistic encoding (i.e corruption) process $q(\mathbf{x}_\tau|\mathbf{x}_0)$, and a corresponding, parametric decoding (i.e generative) process $p_\theta(\mathbf{x}_0|\mathbf{x}_\tau)$. We first roto-translationally align the endpoints of the path via a Kabsch alignment (Kabsch, 1993). We encode the aligned endpoints of the path into the chosen latent level for the initial guess, $\tau_{\text{initial}}$ to produce two latent endpoints $\mathbf{z}^{(0)} \sim q(\mathbf{z}_{\tau_{\text{initial}}}|\mathbf{x}^{(0)})$ and $\mathbf{z}^{(L)} \sim q(\mathbf{z}_{\tau_{\text{initial}}}|\mathbf{x}^{(L)})$. We then interpolate linearly to generate a latent path $\mathbf{Z} = \left\{\mathbf{z}^{(i)} = (1 - \frac{i}{L})\mathbf{z}^{(0)} + \frac{i}{L}\mathbf{z}^{(L)}\right\}_{i\in[0,L]}$. We can then either decode the path back to the configurational space via $p_\theta(\mathbf{x}|\mathbf{z}^{(i)})$ to obtain a path $\mathbf{x}$, in which case the subsequent OM optimization would occur in configurational space, or defer decoding, in which case optimization occurs at the latent level $\tau_{\text{initial}}$ starting from the latent path $\mathbf{Z}$. Intuitively, larger values of $\tau_{\text{initial}}$ produce more diverse initial guesses at the expense of decreasing correspondence to the endpoint states.

---

**Algorithm 2** Initial Guess Path Generation with a Generative Model

---

1:  **Given** a generative model with non-parametric encoder $q$, and decoder $p_\theta$.
2:  **Function** InitialGuess($\mathbf{x}^{(0)}, \mathbf{x}^{(L)}, \tau_{\text{initial}}$)
3:      **Align** both samples via Kabsch alignment:
4:          $\mathbf{x}^{(0)}, \mathbf{x}^{(L)} = \text{KabschAlign}(\mathbf{x}^{(0)}, \mathbf{x}^{(L)})$
5:      **Encode** both samples into latent level $\tau_{\text{initial}}$ of the generative model:
6:          $\mathbf{z}^{(0)} \sim q(\mathbf{z}_{\tau_{\text{initial}}}|\mathbf{x}^{(0)}), \mathbf{z}^{(L)} \sim q(\mathbf{z}_{\tau_{\text{initial}}}|\mathbf{x}^{(L)})$
7:      **Interpolate** linearly (or spherically) in the latent space to generate an initial guess latent path:
8:          $\mathbf{Z} = \left\{\mathbf{z}^{(i)} = (1 - \frac{i}{L})\mathbf{z}^{(0)} + \frac{i}{L}\mathbf{z}^{(L)}\right\}_{i\in[0,L]}$
9:      **Decode** each point on the initial latent path $\mathbf{Z}$ from $\tau_{\text{initial}}$ to $\tau = 0$ to produce a data path:
10:         $\mathbf{x} = \left\{\mathbf{x}^{(i)} \sim p_\theta(\mathbf{x}|\mathbf{z}^{(i)})\right\}_{i\in[0,L]}$
11:     **Return** $\mathbf{x}$

---

When using a classical FF, we do not have access to a generative model. Thus, we must use a different scheme than what is described in 4 to compute the initial guess path. We first start with a small number of replicas in each basin, creating a large gap in the middle of the path. We then optimize with an unphysically large path term, creating a short but interpolating trajectory. After we are satisfied with the initial guess we multiply each replica twice, creating a path of twice the length that we then optimize again. This simple procedure is repeated until we reach desired length of the path. This procedure is described in Algorithm 3.

**Hutchinson Trace Estimator.** The third term in Eq. (14) involves the trace of the Jacobian of the force, $\nabla \cdot F_\theta(\mathbf{x}^{(i)}, \tau_{\text{opt}})$, or equivalently for conservative forces, the trace of the Hessian (Laplacian) of a scalar energy. Naively computing

---

**Algorithm 3** Initial Guess Path Generation with Iterative Unwrapping

---

1: **Function** InitialGuess($\mathbf{x}^{(0)}$,$\mathbf{x}^{(L)}$,$L_1$,$N$)
2: **Initialize** trajectory by copying boundary points $L_1/2$ times on both ends.
3: $\mathbf{x} = \left\{ \mathbf{x}^{(0)},...,\mathbf{x}^{(0)},\mathbf{x}^{(L)},...,\mathbf{x}^{(L)} \right\}$
4: **For** $m$ from 1 to $N$ repeat:
5:     **Duplicate** every point along the path: $\mathbf{x} \leftarrow \left\{ \mathbf{x}^{(0)},\mathbf{x}^{(0)},\mathbf{x}^{(1)},\mathbf{x}^{(1)}...,\mathbf{x}^{(L)}\mathbf{x}^{(L)} \right\}$
6:     **Optimize** Truncated Onsager-Machlup action: $\mathbf{x} \leftarrow \operatorname{argmin}_{\mathbf{x}} S_\theta^{\text{trunc}}(\mathbf{x})$
7: **Obtain** initial guess $\mathbf{x} = \left\{ \mathbf{x}^{(0)},\mathbf{x}^{(1)},...,\mathbf{x}^{(2^N * L_1 - 1)},\mathbf{x}^{(L)} \right\}$

---

gradients of this quantity can be prohibitively expensive. We thus employ the Hutchinson trace estimator (Hutchinson, 1989) to accelerate computation. Formally, let $\mathcal{H}(\mathbf{x}) = \nabla \cdot F_\theta(\mathbf{x},\tau_{\text{opt}}) \in \mathbb{R}^{N_p * d \times N_p * d}$. We approximate the trace of $\mathcal{H}$ as $\operatorname{tr}(\mathcal{H}(\mathbf{x})) \approx \frac{1}{N}\sum_{i=1}^{N} \mathbf{v}^\intercal \mathcal{H}(\mathbf{x})\mathbf{v}$, where $\mathbf{v} \sim \mathcal{N}(0,I)$. By leveraging vector-Jacobian products (VJP), we can compute the trace without materializing $\mathcal{H}$ or its diagonal elements.

In practical terms, the estimator converges rather slowly. Let us denote the approximated trace by $\hat{\operatorname{Tr}}$. One can derive the variance of the estimator as,

$$Var(\hat{\operatorname{Tr}}) = \frac{1}{N}\operatorname{Var}(\mathbf{v}_i \cdot \mathcal{H}(\mathbf{x})\mathbf{v}_i), \tag{99}$$

which, when $\mathbf{v}_i$ are distributed identically means the error of the trace estimator decays as,

$$|\hat{\operatorname{Tr}} - \operatorname{Tr}(\mathcal{H}(\mathbf{x}))| \leq \frac{C}{\sqrt{N}}. \tag{100}$$

Where $C$ depends on the properties of the matrix. This convergence is rather slow and means that one requires many iterations to arrive to an accurate value of the trace. In practice, however, we found that $N = 15$ worked well and led to smooth OM optimization. This is likely due to the fact that our trajectories were composed of many neighboring points that likely had similar Laplacian values.

**Selection of optimization time.** As described in Section 4, the time $\tau_{\text{opt}}$ used to condition the generative model score function $s_\theta(\mathbf{x},\tau_{\text{opt}}$ is treated as a hyperparameter. In principle, $\tau_{\text{opt}} = 0$ ensures maximal correspondence with the true atomistic force field for Boltzmann-distributed data, but consistent with Arts et al. (2023), we find in practice that a small, nonzero value works better. In Fig. 7, we show the average cosine similarity between the true forces and our pretrained denoising diffusion model's score function at various values of $\tau_{\text{opt}}$ for the Müller-Brown and alanine dipeptide systems. Notably, $\tau_{\text{opt}} = 0$ is not the optimal time conditioning with respect to true force recovery for DDPM.

## H. Müller-Brown Potential Experiments

We provide further details on the Müller-Brown experiments in Section 5.1. All experiments were performed on a single NVIDIA RTX A6000 GPU.

**Potential Parameters.** The exact form of the potential used is the following:

$$\begin{aligned} U(x,y) = &-17.3e^{-0.0039(x-48)^2 - 0.0391(y-8)^2} \\ &-8.7e^{-0.0039(x-32)^2 - 0.0391(y-16)^2} \\ &-14.7e^{-0.0254(x-24)^2 + 0.043(x-24)(y-32) - 0.0254(y-32)^2} \\ &+1.3e^{0.00273(x-16)^2 + 0.0023(x-16)(y-24) + 0.00273(y-24)^2} \end{aligned}$$

This generates the potential shown in Fig. 1. Fig. 8 shows OM optimization using the analytical potential as the force field. Increasing the diffusivity yields paths that cross higher energy barriers, aligning with physical intuition. The results with the diffusion model in Section 5.1 align with the paths derived from the analytical potential, confirming the validity of the diffusion model as an approximation of the forces.

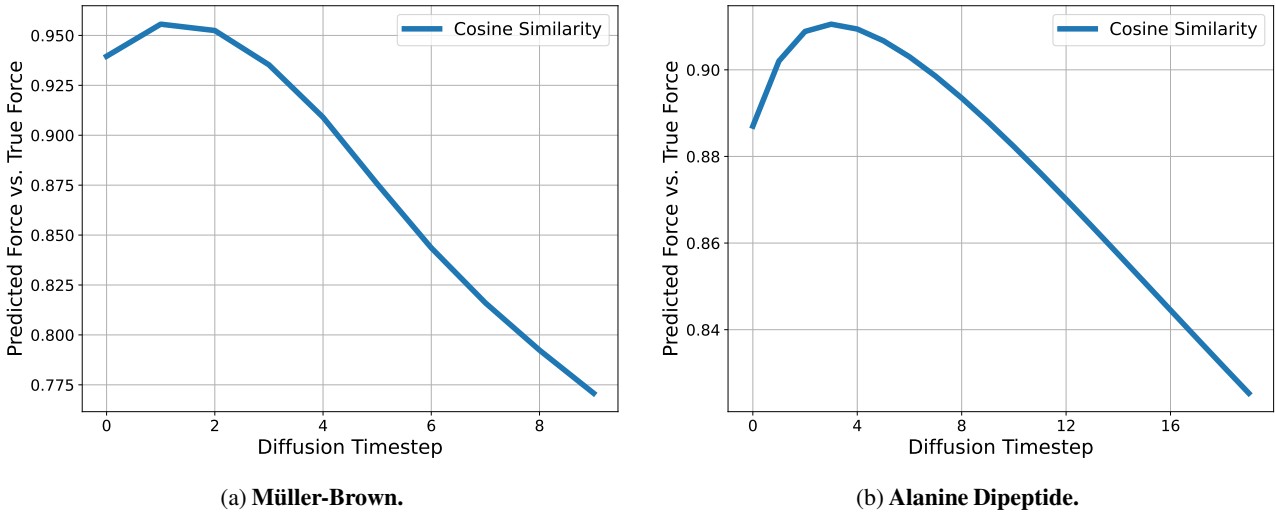

(a) **Müller-Brown.**            (b) **Alanine Dipeptide.**

*Figure 7.* Cosine similarity between true force fields and learned force field from a DDPM's learned time-dependent score model $\mathbf{s}_\theta(\cdot, \tau_{\text{opt}})$ over different diffusion latent times $\tau_{\text{opt}}$. Results are averaged over all paths and atoms, for a Müller-Brown potential and Alanine Dipeptide. Notably, $\tau_{\text{opt}} = 0$ is not the optimal time conditioning with respect to true force recovery for DDPM.

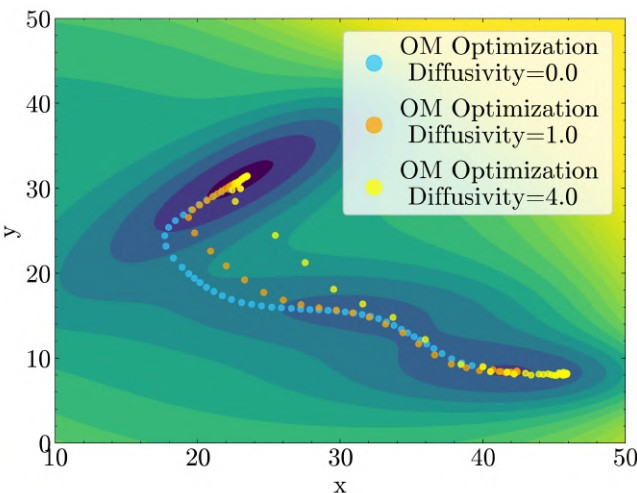

*Figure 8.* **OM optimization can be done with an analytical potential.** We show paths generated with OM optimization using the analytical potential with a timestep of 1, a friction of 1, and multiple diffusivities. Higher diffusivities, corresponding to higher temperatures in physical interpretation, can cross higher energy barriers, aligning with physical intuition.

**Data generation.** To ensure that the transition region is adequately represented with relatively short simulation times, we choose initial conditions for the simulations by uniformly sampling the transition path resulting from OM optimization under the true MB potential. We generate training data by running unbiased, constant-temperature simulations with the MB potential under Langevin dynamics. We run 1,000 parallel simulations for 1,000 steps, yielding a total of 1 million datapoints. Of these, 800,000 are used for training, and 200,000 are reserved for validation.

**Training.** We then train a standard denoising diffusion model on this dataset, with the denoising model parameterized by a 3-layer MLP with a GELU activation (Hendrycks & Gimpel, 2023) and a hidden dimension of 256. The model is trained for 10 epochs, with a batch size of 4096 and a learning rate of $1\mathrm{e}{-3}$ using the Adam optimizer (Kingma & Ba, 2017).

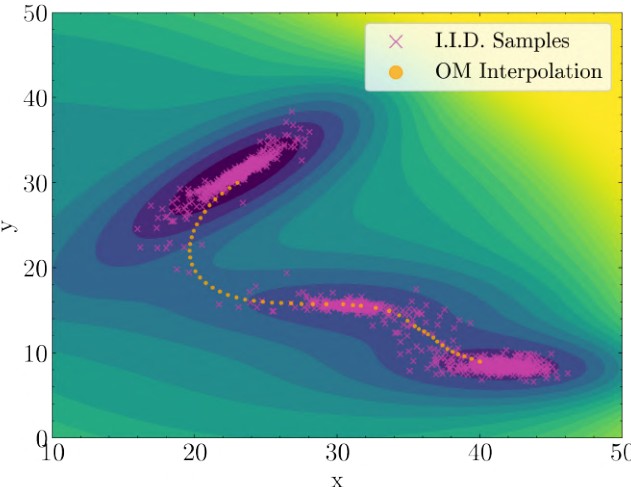

*Figure 9.* **OM optimization from a flow matching model**. The analogous experiment to Fig. 8, but using a flow matching model trained on Müller-Brown data, and using the extracted score via Eq. (17) for OM optimization. Since the stochastic encoding/decoding processes of flow matching models are deterministic and the Müller-Brown setting has a single minimal landscape, the encoding/decoding scheme in itself cannot generate diversity in this setting. The trajectory is generated at diffusivity $D = 0$.

**Energy Laplacian Term.** We estimate the Laplacian of the potential energy surface by using the Hutchinson Trace Estimator (see Section G). As shown in Fig. 10, one random vector ($N = 1$) is enough to capture the local minima and the energy barrier using the Hutchinson Trace Estimator, so we use $N = 1$ in our experiments. Using more random vectors gives a less noisy estimate of the Laplacian, trading off accuracy for computational expense.

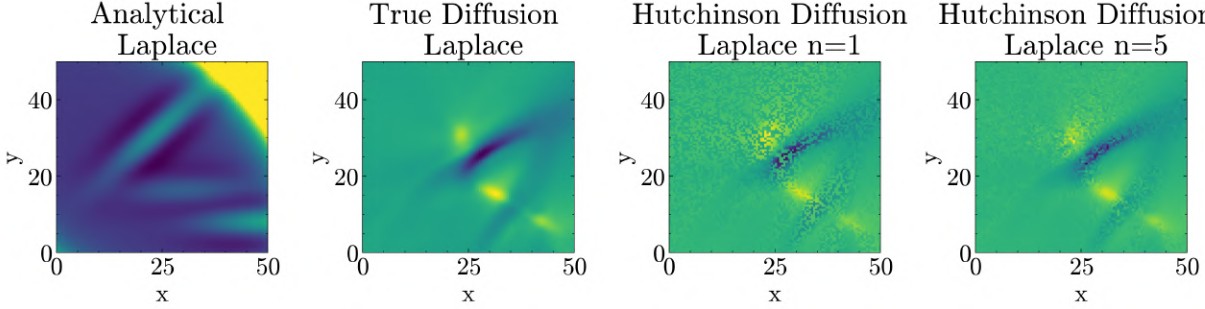

*Figure 10.* **Hutchinson Trace Estimator accurately estimates the Laplacian.** The diffusion model learns an estimate of the Laplacian that captures the Müller-Brown energy wells. The Hutchinson Trace Estimator efficiently approximates this Laplacian, and the estimate becomes less noisy when using more random samples.

**OM Optimization Details.** We pick two points on the potential energy surface (PES) at alternate ends of the transition barrier as target points for interpolation. The hyperparameters for the OM optimization are given in Table 3

**Committor Function and Rate Estimation** We provide additional details on the committor function and transition rate estimation experiment presented in Section 5.1.

According to transition path theory (Vanden-Eijnden et al., 2006), for a transition event between endpoints $\mathcal{A}, \mathcal{B} \in \Omega$, the committor function $q(\mathbf{x})$, captures the probability that a trajectory initiated at $\mathbf{x}_0 = \mathbf{x}$ reaches $\mathcal{B}$ before $\mathcal{A}$:

$$q(\mathbf{x}) = \mathbb{E}[h_{\mathcal{B}}(\mathbf{x}_\tau) \,|\, \mathbf{x}_0 = \mathbf{x}]; \quad \tau = \underset{t \in [0, +\infty)}{\operatorname{argmin}} \quad \{\mathbf{x}_t \in \mathcal{A} \cup \mathcal{B} : \mathbf{x}_0 = \mathbf{x}\}, \tag{101}$$

where $h_{\mathcal{B}}$ is the indicator function for reaching state $\mathcal{B}$. The transition state ensemble is formally defined as the level set

*Table 3.* Hyperparameters used for OM action optimization on the Müller-Brown potential with diffusion.

| Hyperparameter | Value |
|---|---|
| Number of Generated Paths | 50 |
| Action Type | Full |
| Initial Guess Time ($\tau_{\text{initial}}$) | 8 |
| Optimization Time ($\tau_{\text{opt}}$) | 8 |
| Optimization Steps | 200 |
| Optimizer | Adam |
| Learning Rate | 0.2 |
| Path Length ($L$) | 50 |
| Action Timestep ($\Delta t$) | 0.01 |
| Action Friction ($\gamma$) | 0.01 |
| Action Diffusivity ($D$) | 0, 1.0, 4.0 |

$\{\mathbf{x} \in \Omega : q(\mathbf{x}) = 0.5\}$. The committor function $q(\mathbf{x})$ can be used to estimate transition rates between a reactant state $\mathcal{A}$ and product state $\mathcal{B}$ using the following formula (Vanden-Eijnden et al., 2006):

$$k = \frac{k_B T}{\gamma} \int_{\mathbf{x} \in \Omega} p(\mathbf{x}) |\nabla_{\mathbf{x}} q(\mathbf{x})|^2 dx = \frac{k_B T}{\gamma} \langle |\nabla_{\mathbf{x}} q(\mathbf{x})|^2 \rangle_{\Omega}, \tag{102}$$

where $p(\mathbf{x})$ is the probability density under the Boltzmann distribution and $\Omega$ is the configuration space. Thus, with a differentiable estimate of the committor function, we can compute reaction rates as an ensemble average over samples from $p(\mathbf{x})$. By applying the Backward Kolmogorov Equation (BKE) and Vainberg's Theorem, we can show that the committor function is the solution to the following functional optimization problem (Hasyim et al., 2022):

$$q(\mathbf{x}) = \operatorname*{argmin}_{\widetilde{q}} \mathcal{L}[\widetilde{q}] = \operatorname*{argmin}_{\widetilde{q}} \frac{1}{2} \langle |\nabla_{\mathbf{x}} \widetilde{q}(\mathbf{x})|^2 \rangle_{\Omega \setminus \mathcal{A} \cup \mathcal{B}}, \tag{103}$$

subject to the boundary conditions $\widetilde{q}(\mathbf{x}) = 0, \mathbf{x} \in \partial \mathcal{A}; \widetilde{q}(\mathbf{x}) = 1, x \in \partial \mathcal{B}$. Thus, we can train a neural network approximation to the committor function $q_\theta(\mathbf{x})$ by extremizing the following loss function:

$$L(\theta) = \frac{1}{2} \langle |\nabla_{\mathbf{x}} q_\theta(\mathbf{x})|^2 \rangle_{\Omega \setminus \mathcal{A} \cup \mathcal{B}} + \lambda_{\mathcal{A}} \frac{1}{2} \langle q_\theta(\mathbf{x})^2 \rangle_{\mathcal{A}} + \lambda_{\mathcal{B}} \frac{1}{2} \langle (1 - q_\theta(\mathbf{x}))^2 \rangle_{\mathcal{B}} \tag{104}$$

The first term minimizes the BKE functional, and the second and third terms enforce the boundary conditions with penalty strengths $\lambda_{\mathcal{A}}$ and $\lambda_{\mathcal{B}}$, respectively. The ensemble averages in the loss, which are high-dimensional integrals whose computational cost grows exponentially with system size, are in practice by replaced by importance-sampled Monte Carlo averages over a dataset of samples $\mathcal{D}_{\text{train}} = \{x_i\}_{i=1}^{N}$ from MD simulations, MCMC, or any enhanced sampling method (such as OM optimization). Once trained, the neural network committor function $q_\theta(\mathbf{x})$ can be used to estimate the rate via Eqn 102, substituting $q_\theta(\mathbf{x})$ in place of $q(\mathbf{x})$, and averaging over $\mathcal{D}_{\text{train}}$ instead of the intractable full ensemble average over $\Omega$.

$$k_\theta = \frac{k_B T}{\gamma} \langle |\nabla_{\mathbf{x}} q_\theta(\mathbf{x})|^2 \rangle_{\Omega} \approx \hat{\mathbb{E}}_{\mathbf{x} \sim \mathcal{D}_{\text{train}}} |\nabla_{\mathbf{x}} q_\theta(\mathbf{x})|^2 \tag{105}$$

Notice that the neural network estimate of the rate $k_\theta$ is equivalent (up to constants) to the BKE functional loss $\mathcal{L}[\widetilde{q}]$. The primary challenge with learning the committor function in this way is the issue of sampling: the optimization problem is dominated by rare events with large values of $|\nabla_{\mathbf{x}} q_\theta(\mathbf{x})|^2$ (i.e events in the transition region). Therefore, if the transition region(s) are insufficiently sampled, the training procedure will likely fail to converge, and the estimate rate $k_\theta$ will be inaccurate. Therefore, committor and rate estimation using this method reduces to a rare event sampling problem. Inspired by Hasyim et al. (2022), our idea is to use the transition paths obtained from our OM optimization procedure (shown in Fig. 11a) as samples over which to minimize Eq. (104). Specifically, for the Müller-Brown potential, we first sampled more points by initiating unbiased Langevin dynamics simulations along the OM-optimized paths (100 simulations, each 50 ps long, $k_B T = 1$), using the diffusion model's score function at $\tau = 0$ as the force field (we could have also used the analytical forces from the Müller-Brown potential). This resulted in a collection of points $\mathcal{D}_{\text{train}} = \{x_i\}_{i=1}^{N}$ (shown in Fig. 11b). To account for the biased initial conditions of the simulations, we reweight the samples in $\mathcal{D}_{\text{train}}$ to the underlying Boltzmann distribution using the reweighting factors

$$w_i = \frac{\frac{e^{-\beta U(x_i)}}{p_{\text{OM}}(x_i)}}{\sum_i \frac{e^{-\beta U(x_i)}}{p_{\text{OM}}(x_i)}}, \tag{106}$$

where $\beta = \frac{1}{k_B T}$, $U(\mathbf{x})$ is the analytical Müller-Brown potential energy, and $p_{OM}$ is the empirical density of $\mathcal{D}_{\text{train}}$, obtained via binning. We train a 5-layer MLP with a hidden dimension of 64 and a sigmoid activation function to approximate the committor function. The model is trained for 2000 steps with a batch size of 4096, using an Adam optimizer with a learning rate of 0.0001 and a cosine decay scheduler. The boundary condition loss weights are $\lambda_{\mathcal{A}} = \lambda_{\mathcal{B}} = 20$. The final estimated committor function profile is shown in Fig. 11c. We obtain a transition rate estimate of $1.3 \times 10^{-5}$, compared with the true rate of $5.4 \times 10^{-5}$ obtained by solving the Backward Kolmogorov Equation numerically using a finite-element method in FEniCS (Baratta et al., 2023).

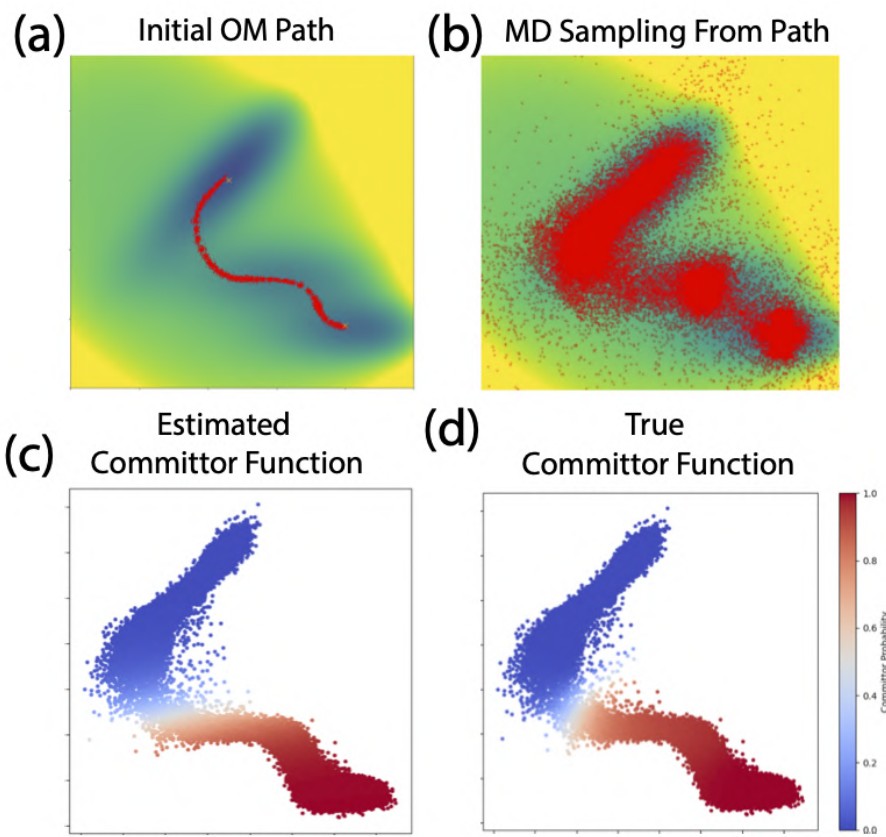

*Figure 11.* **Efficient committor and transition rate estimation on the 2D Müller-Brown potential using OM optimization. (a)** Initial OM optimized transition path with a pretrained denoising diffusion model, using $\tau_{\text{opt}} = 4$ and 100 paths. **(b)** Subsequent diversification of the sampled transition paths by performing unbiased Langevin dynamics simulations initiated along the OM optimized paths (100 simulations, each 10,000 steps, $k_B T = 1$) with the diffusion model's score function at $\tau = 0$. **(c)** Predicted values of the committor function $q(\mathbf{x})$, obtained via minimizing $\langle |\nabla_{\mathbf{x}} q_\theta(\mathbf{x})|^2 \rangle_\Omega$, where $\langle \cdot \rangle_\Omega$ denotes an ensemble average over the samples collected in part b) and the committor function $q_\theta(\mathbf{x})$ is parameterized by a 5 layer MLP with hidden dimension of 64 and sigmoid activation. **(d)** True committor function calculated by solving the Backward Kolmogorov Equation numerically using a finite-element method in FEniCS. The reaction rate $k$ is computed via the relation $k = \frac{k_B T}{\gamma} \langle |\nabla_{\mathbf{x}} q(\mathbf{x})|^2 \rangle_\Omega$. Our predicted reaction rate is $1.3 \times 10^{-5}$, compared with the true rate of $5.38 \times 10^{-5}$.

## I. Alanine Dipeptide Experiments.

We present additional results on the alanine dipeptide results in Section 5.2.

**Details on Traditional Enhanced Sampling Baselines.** For MCMC, we report metrics found in (Du et al., 2024) for the fixed-length, two-way, exact method. We implement metadynamics ourselves with Ramachadran angles as CVs. We use a hill height of 0.2kcal/mol, a width of 10Å, and a temperature of 300K. We report the number of force field (or generative model score) evaluations required per 1,000 step transition path (a batched computation over many samples is considered to be one evaluation).

**Diffusion Model Training.** We construct a training dataset from MD simulations of alanine dipeptide at 1000K (a large temperature is chosen to promote the inclusion of transition regions in the data). We then train a diffusion model on these samples, using the Graph Transformer architecture described in Section J. We train models for 50,000 steps, using an Adam optimizer with a learning rate of 0.0004 and a cosine annealing schedule reducing to a minimum learning rate of 0.00001. We use an exponential moving average with $\alpha = 0.995$. The diffusion model uses 1,000 integration steps at inference time.

**OM Optimization Details.** We choose a path length of $L = 300$ and noise/denoise by 300 steps (out of 1,000) to initialize via Algorithm 2, then conduct OM optimization for 50,000 steps, fixing latent diffusion time $\tau_{\text{opt}} = 3$. $\gamma$ and $\Delta t$ are set to 0.7 and 1e-4 respectively. After OM optimization, we run a short energy minimization procedure for 1400 steps using the Amber *ff19SB* classical force-field to relax structures, followed by OM optimization using BFGS algorithm that required around 21,000 steps.

**Calculating Free Energy Barriers with Umbrella Sampling.** Since we optimized with the truncated OM action Eq. (4) we obtained minimum energy paths. To compute free energy barriers that are necessary for reaction rate predictions and comparisons with literature, we employ some sampling to estimate the entropic contribution. On alanine dipeptide, we use the following procedure to obtain free energy profiles along the OM-optimized paths.

1. Perform umbrella sampling (Torrie & Valleau, 1977): short, 10ps, simulations with a harmonic string potential centered around the points that define the original minimum energy path. The spring constant used had value of 5 kcal/mol/rad$^2$

2. Fit a collective variable (CV) on the resulting samples. We choose Principal Component Analysis (PCA).

3. Employ the WHAM algorithm (Kumar et al., 1992) to obtain the reweighted potential of mean force, shown in Fig. 3b.

## J. Fast-Folding Protein Experiments

We present additional details on the fast-folding protein results in Section 5.3. All experiments were performed on a single NVIDIA RTX A6000 GPU.

**MD Simulations with Diffusion Model** To assess the computational efficiency and accuracy of our OM optimization relative to alternative methods of sampling the configurational space (see Fig. 4a and b), we run Langevin MD simulations for varying lengths of time (up to 12 ns) using the diffusion model's score function as an effective force field (Arts et al., 2023), with a timestep of 2 fs. To ensure a fair comparison, we set the number of parallel MD trajectories to be the same as the number of transition paths sampled with OM optimization.

**Markov State Model Construction.** We provide further details on the Markov State Model analysis used to evaluate the quality of transition paths for the fast-folding protein experiments. We largely follow the procedure described in Jing et al. (2024b).

We perform k-means clustering of the reference MD simulations into 20 clusters using the top 2 Time Independent Component (TIC) dimensions, which are fit on the pairwise distances and dihedral angles of the protein configurations. We then fit a Markov State Model (MSM) with a lagtime of 200ps (the frequency at which the simulations were saved) to obtain a transition probability matrix $T$ between the 20 discrete states in the MSM (e.g, $T_{j,k} = p(s_{t+1} = k | s_t = j)$, where $s_t$ and $s_{t+1}$ are the states at time $t$ and $t+1$). This constitutes the reference MSM.

To evaluate transition paths sampled from our OM optimization method, we first discretize them under the reference MSM (i.e represent them as a sequence of cluster indices between 1 and 20). We subsample the paths to be of length 20. From this, we compute the probability of the path under the reference MSM via:

We also sample 1,000 discrete, reference paths of length $L = 20$ (corresponding to a transition time of 200ps $\times$ 20 = 4ns) from the reference MSM, conditioned on the start and end states $s_1$ and $s_L$ (these are the cluster indices of the transition endpoints $\mathbf{x}^{(0)}$ and $\mathbf{x}^{(L)}$). This can be achieved by sampling states $s_2 ... s_{19}$ iteratively as

$$s_{t+1} \sim \frac{T^{(L-t-1)}_{:,s_L} T_{s_L,:}}{T^{(L-t)}_{s_t, s_L}}, \tag{107}$$

where the superscipt denotes a matrix exponential. See Jing et al. (2024b) for precise details.

With both the reference and generated discretized paths, we compute the following metrics:

1. **Jensen-Shannon Divergence.** Consider the probability of each MSM state based on the frequency at which it is visited in the discretized paths. We compute these probabilities for both the reference and generated paths, and compute the JSD between the resulting categorical distributions.

2. **Transition Negative Log Likelihood.** The average negative log likelihood of a transition from one discretized MSM state to the next, averaged over transitions from all generated paths with nonzero probability under the reference MSM. Since there are $L-1$ individual transitions for a path of length $L$, under the Markovian assumption the average negative log likelihood of a transition is given by $-\frac{1}{L-1}\log P(s_1...s_L) = -\frac{1}{L-1}\sum_{t=1}^{L-1}\log\left(\frac{T_{s_t,s_L}^{(L-t-1)}\cdot T_{s_t,s_{t+1}}}{T_{s_t,s_L}^{(L-t)}}\right).$

3. **Fraction of Valid Paths.** The fraction of generated paths with nonzero probability under the reference MSM.

When considering replicate MD simulations of different lengths (e.g 1 $\mu$s, 10 $\mu$s), we fit a MSM to the simulations using the same discretized clusters as were used to fit the reference MSM, and sample 1,000 paths of length $L=20$ in the same way described above.

We note that the reported metrics are sensitive to the choice of the MSM trajectory length $L$ used for the reference simulations. Varying $L$ changes the time horizon of the transition and yields qualitatively different reference paths. Since we chose $L=20$, the reported metrics reflect the correspondence of generated paths with true paths of length 4 ns. We also find that in practice, the optimal choice of $\Delta t$ in the OM action (i.e., the choice that yields the best MSM metrics as defined above) results in time horizons $T_p = L\Delta t$ (where $L$ here is the number of path discretization points during OM optimization) considerably smaller than the reference time horizon of 4 ns (e.g., for the typical choices of $L=200$, $\Delta t = 1fs$, we have $T_p = 0.2ps$). In other words, OM optimization yields accelerated transition dynamics, because it largely bypasses local/minor fluctuations that would occur in an actual MD simulation.

**Committor Function Analysis.** The committor function $q(\mathbf{x})$ (see Section 5.1 for definition) is obtainable as the solution to the steady-state backward Kolmogorov equation (BKE) (Hasyim et al., 2022), which is generally infeasible to solve directly or numerically for high-dimensional systems. For the fast-folding proteins, we obtain an empirical estimate of the committor function by dividing the TIC configuration space of each protein into $100^2$ discrete bins, and replacing the expectation in Eq. (101) with an empirical average over trajectories starting from each bin in the reference MD simulations from Lindorff-Larsen et al. (2011). The resulting committor estimates for the fast-folding proteins are shown in Fig. 12.

**Model Architecture and Training.** Our denoising diffusion and flow matching generative models are parameterized by a Graph Transformer architecture identical to what was used in Arts et al. (2023) (in the case of diffusion, we use the exact pretrained model from Arts et al. (2023)). To summarize, nodes are featurized by the ordering of each residue in the overall sequence, while edges are featurized by the pairwise $C_\alpha$-$C_\alpha$ distances. Nodes and edges are then jointly treated as tokens for input to the Transformer, which updates the token representations. A scalar output is obtained by summing learned linear projections of the token representations. Both the denoising diffusion vector field $\epsilon_\theta$ and the flow model velocity field $v_\theta$ are parameterized as the gradient of the final scalar output of the model with respect to the input $C_\alpha$ coordinates.

For denoising diffusion, we use the pretrained models from Arts et al. (2023). For flow matching, we train our own models. We train models with 3 attention layers for 1 million iterations, using an Adam optimizer with a learning rate of 0.0004 and a cosine annealing schedule reducing to a minimum learning rate of 0.00001. We use an exponential moving average with $\alpha = 0.995$. The diffusion models use 1,000 integration steps at inference time, while the flow matching models use 10 steps.

Protein-specific training and architecture hyperparameters are given in Table 4.

*Table 4.* Architecture and training hyperparameters for diffusion and flow matching generative models on fast-folding proteins.

| Hyperparameter | Chignolin | Trp-cage | BBA | Villin | Protein G |
|---|---|---|---|---|---|
| Batch size | 512 | 512 | 512 | 512 | 256 |
| Number of hidden features | 64 | 128 | 96 | 128 | 128 |

**OM Optimization Details.** We list all the optimization hyperparameters used to perform OM optimization on the fast-folding proteins, for both diffusion and flow matching models, in Tables 5 and 6.

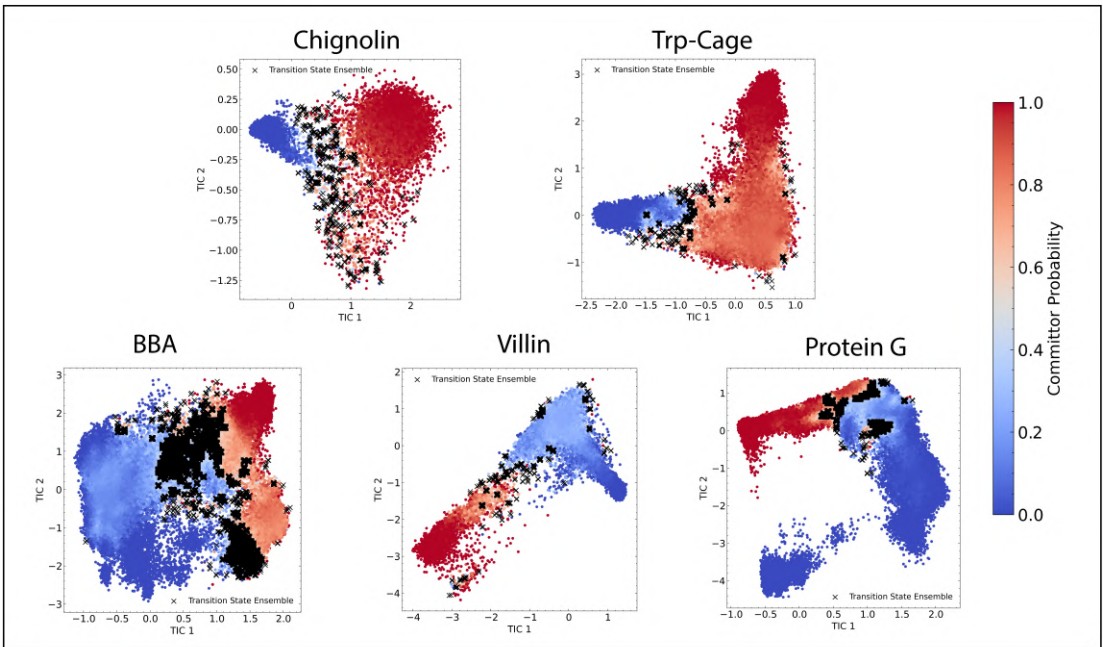

*Figure 12.* **Empirical committor landscapes for fast-folding proteins.** The committor is computed by binning the conformational space into $100^2$ bins and measuring the frequency at which reference MD trajectories initiated in each bin reach the target state before the start state. The empirical transition ensemble (shown in black) is defined as the level set $\{\mathbf{x} : 0.45 \leq q(\mathbf{x}) \leq 0.55\}$.

*Table 5.* Hyperparameters used for OM action optimization on fast-folding proteins with diffusion models.

| Hyperparameter | Chignolin | Trp-cage | BBA | Villin | Protein G |
|---|---|---|---|---|---|
| Number of Generated Paths | 8 | 8 | 32 | 4 | 4 |
| Action Type | Truncated | Truncated | Full | Truncated | Truncated |
| Initial Guess Time ($\tau_{\text{initial}}$) | 250 | 250 | 250 | 250 | 250 |
| Optimization Time ($\tau_{\text{opt}}$) | 20 | 15 | 20 | 10 | 10 |
| Optimization Steps | 5000 | 5000 | 5000 | 5000 | 5000 |
| Optimizer | Adam | Adam | SGD | SGD | SGD |
| Learning Rate | 0.2 | 0.2 | 1e-5 | 1e-5 | 1e-5 |
| Path Length ($L$) | 200 | 200 | 200 | 200 | 200 |
| Action Timestep ($\Delta t$) | 0.001 | 0.001 | 0.001 | 0.005 | 0.002 |
| Action Friction ($\gamma$) | 1 | 1 | 1 | 1 | 1 |
| Action Diffusivity ($D$) | 0 | 0 | 1 | 0 | 0 |

**Comparison to Transition Paths from Reference MD Simulations.** We directly compare our generated paths against the unbiased MD transition trajectories from the D.E.Shaw reference simulations (Lindorff-Larsen et al., 2011), finding that our paths sample the correct regions of phase space (Fig. 13).

**Visualization of Transition Paths.** In Figures 14, 15, and 16, we provide additional visualizations of transition paths sampled by our diffusion and flow matching models for the fast folding proteins, both in TIC and atomic space.

**Transition Paths with Varying Time Horizons** We vary the time horizon $T_p$ over which OM optimization is performed, and increase the timestep $\Delta t$ to maintain the same number of discretization points $L$ for computational efficiency. As shown in Fig. 17, paths with varying time horizons explore different parts of the transition state ensemble (black), with longer horizon paths spending more time near the bottom-right energy well corresponding to the folded state, and shorter paths taking more direct paths between the endpoints.

**Training without Transition Regions.** We provide additional details on the data-starved experiment described in Section 5.3. Transition states are challenging to sample, and therefore may not be abundant in reference MD simulations or structural

*Table 6.* Hyperparameters used for OM action optimization on fast-folding proteins with flow matching models.

| Hyperparameter | Chignolin | Trp-cage | BBA | Villin | Protein G |
|---|---|---|---|---|---|
| Number of Generated Paths | 8 | 8 | 32 | 4 | 4 |
| Action Type | Truncated | Truncated | Full | Truncated | Truncated |
| Initial Guess Time ($\tau_{initial}$) | 7 | 7 | 7 | 7 | 7 |
| Optimization Time ($\tau_{opt}$) | 0.5 | 0.5 | 0.5 | 0.5 | 0.5 |
| Optimization Steps | 5000 | 5000 | 5000 | 5000 | 5000 |
| Optimizer | Adam | Adam | SGD | SGD | SGD |
| Learning Rate | 0.2 | 0.2 | 1e-4 | 1e-4 | 1e-4 |
| Path Length ($L$) | 200 | 200 | 200 | 200 | 200 |
| Action Timestep ($\Delta t$) | 0.0008 | 0.0005 | 0.001 | 0.001 | 0.001 |
| Action Friction ($\gamma$) | 1 | 1 | 1 | 1 | 1 |
| Action Diffusivity ($D$) | 0 | 0 | 1 | 0 | 0 |

databases, which typically serve as training datasets for generative models. To simulate the scenario in which the underlying dataset is not exhaustive and under-represents the rare, transition regions, we remove 99% of the datapoints for which $0.1 \leq q(\mathbf{x}) \leq 0.9$, where $q(\mathbf{x})$ is the empirical committor value (described in **Committor Function Analysis**). Thus, most of the remaining datapoints have committor values close to 0 or 1, meaning they initiate trajectories which stay in their respective local energy minima without transitioning across the path. For Chignolin, Trp-Cage, and BBA, the subsampling procedure removes 1.4%, 68%, and 85% of the datapoints, respectively. We train diffusion models on these subsampled datasets using the same hyperparameters described earlier, followed by OM optimization between the same endpoints, using the same hyperparameters as used before. As shown in Fig. 18, the produced transition paths are similar to those shown in Figures 4a and 16. The paths still pass through the expected transition state regions (denoted in black), despite having seen them at a much lower frequency during training.

## K. Tetrapeptide Experiments

We provide further details on the tetrapeptide experiments from Section 5.4. All experiments were performed on a single NVIDIA RTX A6000 GPU.

**Heavy-Atom Representation.** Following Jing et al. (2024b), we use a heavy-atom representation of the tetrapeptides (that is, hydrogens are excluded from the representation). The terminal oxygen (OXT) of the C-terminus is also excluded. This results in tetrapeptides with at most 56 atoms each.

**Training.** We train a flow matching model, parameterized by a Graph Transformer with all the same architecture and training hyperparameters used for the fast-folding proteins (Section J), with the only differences being the inclusion of learnable atom type embeddings and the use of padding tokens due to the variable number of atoms in each tetrapeptide. The atom embeddings are concatenated to the residue ordering and the flow timestep to form the node features. We train on a dataset of 3109 tetrapeptides simulated in (Jing et al., 2024b), taking 10,000 evenly spaced configurations from the simulations for each tetrapeptide. We use the same optimizer and learning rate scheduler as with the fast folding proteins (J), training for 100,000 steps with a batch size of 2048.

**OM Optimization on Held-Out Proteins.** We perform OM optimization on 100 held-out tetrapeptide sequences not seen during training (using the same splits as in (Jing et al., 2024b)). As in Jing et al. (2024b), we choose endpoints for the transition paths by picking the pair of states from the MSM with the lowest, non-zero probability flux between them, ensuring that the transitions are challenging but feasible to find.

The optimization hyperparameters are given in Table 7.

**Energy Minimization.** After OM optimization, we add the missing OXT and hydrogen atoms (at neutral pH) back to the structures using PDBFixer (`https://github.com/openmm/pdbfixer`). We then perform up to 200 steps of energy minimization in OpenMM in vacuum with the amber14 force field using L-BFGS. We restrict the Kabsch-aligned RMSD between the initial and final structures to 1 Angstrom to ensure that only very minor structural changes occur.

**Evaluation.** We use the same Markov State Model-based evaluation pipeline described in Section J for the fast-folding proteins. Following (Jing et al., 2024b), the TIC dimensions are fit on the backbone and sidechain torsion angles. The reported

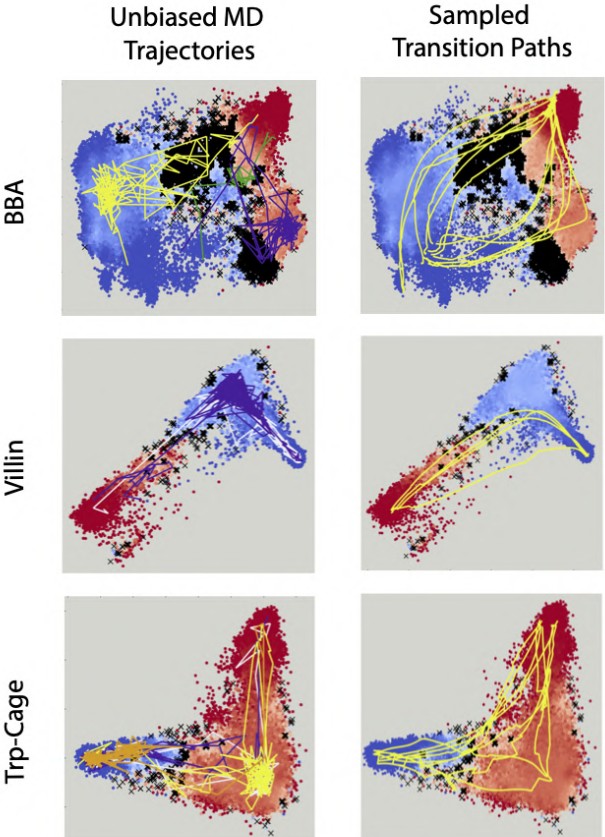

*Figure 13.* Sample trajectories from reference, unbiased fast-folding protein MD simulations from D.E.Shaw (left) and paths sampled via OM optimization (right) overlaid on TICA plots with committor values shaded from blue to red. We observe strong agreement in the TICA space, with our OM paths traversing the same transition regions and exhibiting similar structural transitions. While the OM paths are smoother/do not contain the same fluctuations found in unbiased MD trajectories, they capture the same essential features. OM optimization somewhat oversamples transition paths which avoid energy wells. This could be improved by increasing the time horizon of the transition to accommodate time spent in kinetic traps.

metrics are averaged over the 100 held-out test proteins.

**Visualization of Sampled Paths.** We provide TIC visualizations of sampled transition paths for selected tetrapeptides alongside paths from the reference MSM in Fig. 19.

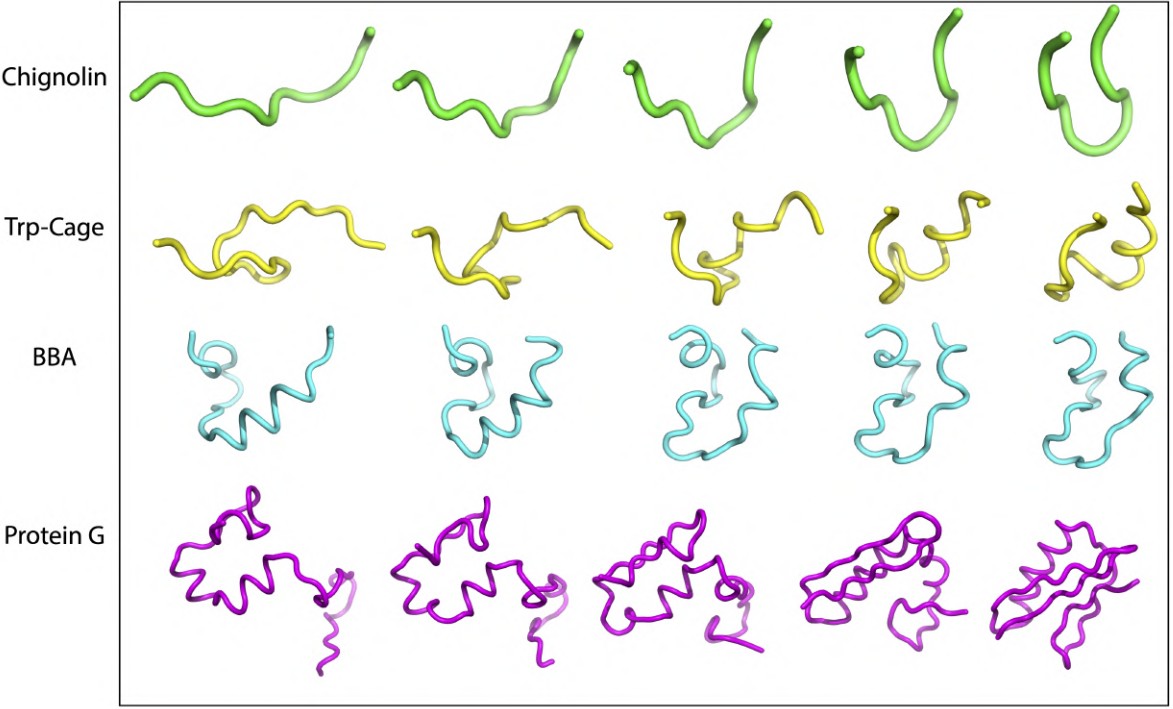

*Figure 14.* Visualization of sampled transition paths from OM optimization with pretrained diffusion models for fast-folding proteins.

*Table 7.* Hyperparameters used for OM action optimization on tetrapeptides with flow matching models.

| Hyperparameter | Value |
|---|---|
| Number of Generated Paths | 16 |
| Action Type | Truncated |
| Initial Guess Time | 7 |
| Optimization Time | 0.5 |
| Optimization Steps | 250 |
| Optimizer | Adam |
| Learning Rate | 0.2 |
| Path Length ($L$) | 100 |
| Action Timestep ($\Delta t$) | 0.0001 |
| Action Friction ($\gamma$) | 1 |
| Action Diffusivity ($D$) | 0 |

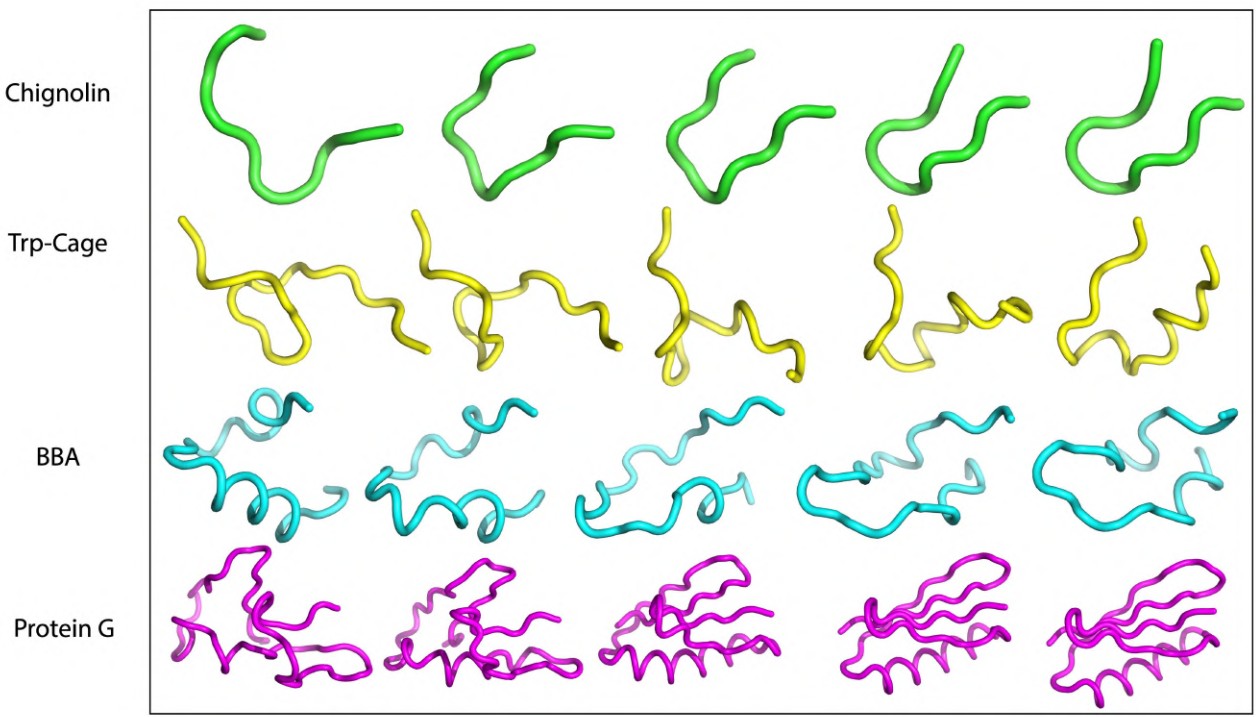

*Figure 15.* Visualization of sampled transition paths from OM optimization with pretrained flow matching models for fast-folding proteins.

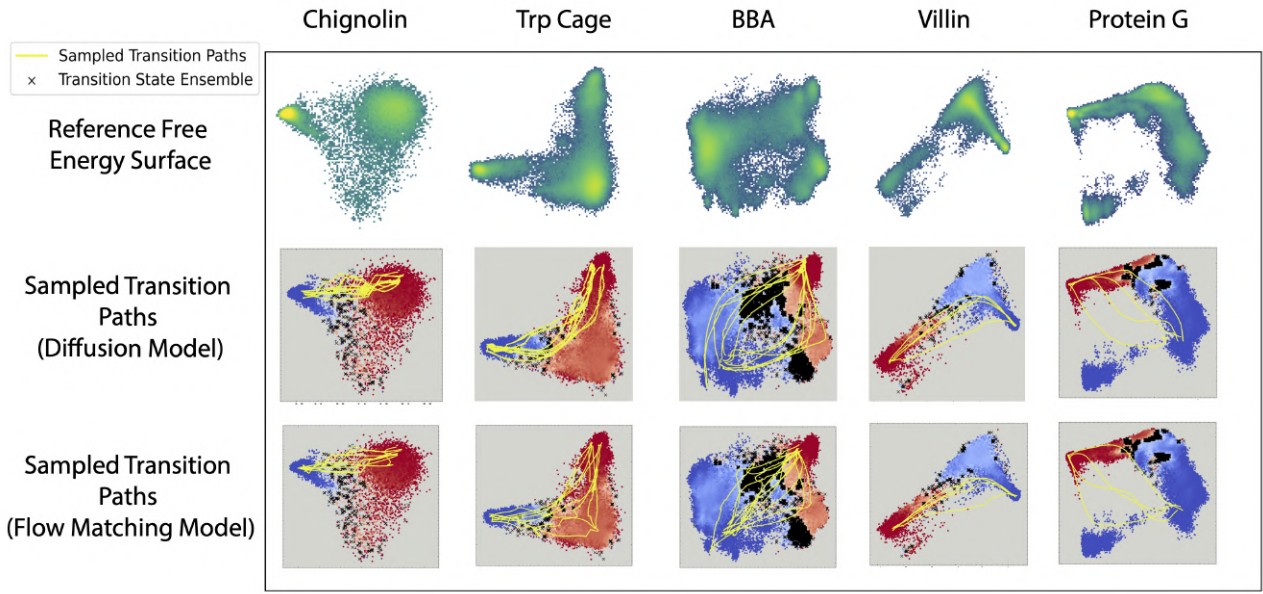

*Figure 16.* Visualization of sampled fast-folding protein transition paths in TIC space.

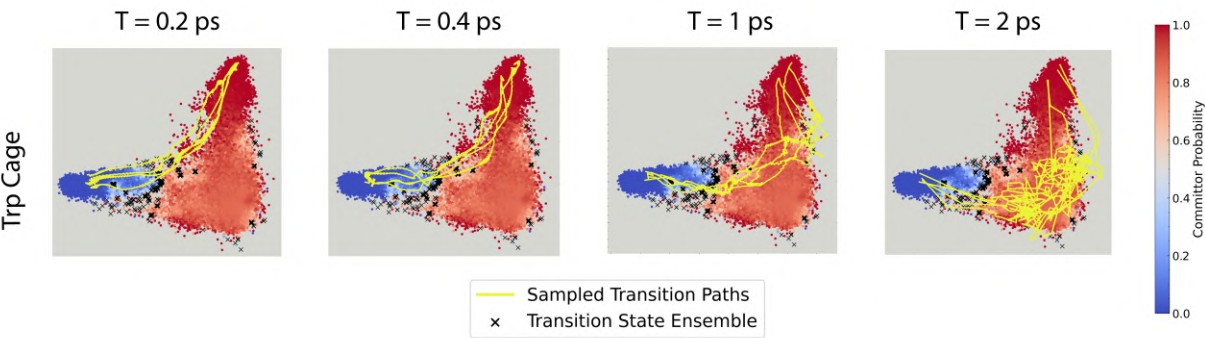

*Figure 17.* **Varying time horizons of path sampling.** OM-optimized transition paths for Trp-Cage using physical values for hyperparameters controlling relative contributions of path and force norm terms in the OM action. Here we set $\gamma = 1\mathrm{ps}^{-1}$, and vary the transition time horizon over $T_p \in \{0.2, 0.4, 1, 2\}$ ps and the timestep over $\Delta t \in \{0.001, 0.002, 0.005, 0.01\}$ ps, which fixes $L = 200$ discretization points.

*Figure 18.* **Training datasets and sampled transition paths resulting from removing intermediate committor function values. (Top Row)** Original training datasets. **Middle Row** Datasets resulting from removing 99% of datapoints with committor values (obtained empirically) between 0.1 and 0.9. **Bottom row.** Transition paths resulting from OM optimization with a diffusion model trained on the subsampled datasets.

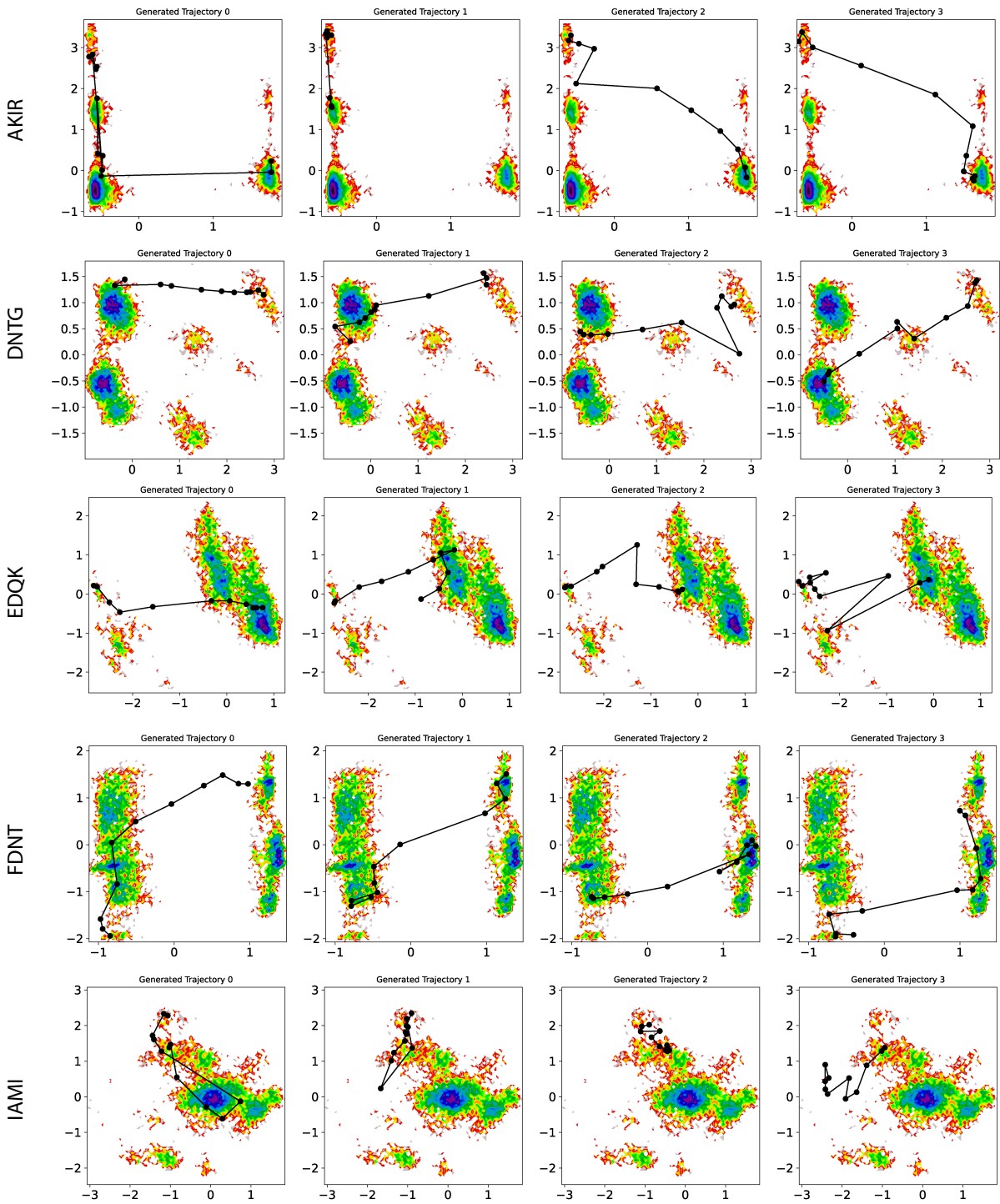

*Figure 19.* **Sampled transition paths from OM optimization on selected, held-out tetrapeptide sequences.** The sampled paths are diverse and intuitively pass through high density regions in the TIC free energy landscape.

