# OpenReview forum: "Action-Minimization Meets Generative Modeling: Efficient Transition Path Sampling with the Onsager-Machlup Functional"
_ICML.cc/2025/Conference — ICML 2025 poster_

### Official Review · Reviewer_bRih · 2025-02-21

**Overall Recommendation:** 3

**Summary:**

This paper presents a new method for transition path sampling in molecular systems by combining generative models with the Onsager-Machlup action functional. The authors show how pre-trained generative models (specifically denoising diffusion and flow matching) can be repurposed to find high-probability transition paths between stable configurations without requiring specialized training. The author's approach interprets candidate paths as trajectories sampled from stochastic dynamics induced by the learned score function, making finding transition paths equivalent to minimizing the Onsager-Machlup action. The method is demonstrated on a 2D Müller-Brown potential, fast-folding proteins, and tetrapeptide sequences, showing it can generate physically realistic transition pathways. Compared to traditional molecular dynamics simulations, their approach produces higher-quality transition paths with significantly less computational cost.

**Claims And Evidence:**

The overall claims focus on the efficiency and physically realistic transition path generated based on the OM algorithm, supported by its implementation on systems such as Müller-Brown potential and coarse-grained proteins.

Comparisons are made between OM-generated paths and MD simulations in a few systems with wall-clock time comparison presented in Figure 3c shows that OM optimization is faster than brute-force MD. However, the comparison lacks benchmarking against enhanced sampling methods (e.g., metadynamics, umbrella sampling, weighted ensemble, transition interface sampling). These are the actual competitors, not brute-force MD. Therefore, The computational cost of optimizing the OM action versus performing biased MD simulations is not clearly reported.

Regarding the generated transition path, the claim of its physical realism requires additional proof. The paper claims that minimizing the OM action produces physically meaningful transition paths, even for systems unseen during training, given the comparison between transition path and free energy landscapes. Since the generative models are trained on equilibrium data, the core issue is that the score-based generative models do not inherently capture rare transition pathways. The paths generated may be interpolations that resemble plausible transitions rather than actual dynamical pathways governed by the correct physics (figure 13 and 14). Besides, MSM-based validation is useful but does not confirm the dynamical accuracy of the generated paths. The MSM itself is only as good as the data used to construct it.

Another claim that requires further examination is the zero-shot generalization statement. However results on held-out tetrapeptides suggest that transition paths can be generated for new sequences, different tetrapeptides have unique energy landscapes, and it is unclear how well a generative model trained on one set of peptides generalizes to another with different interaction potentials. Generative models trained on equilibrium data are likely to underrepresent high-energy, transition-state conformations, which means the model might struggle with generating physically accurate rare event dynamics.

The strong claim of solving the CV selection problem is not solid enough -- the method optimizes transition paths based on the OM action but does not guarantee exploration of the full transition ensemble. If the generative model was trained on biased datasets (e.g., experimental structures, short MD trajectories, not-complete stationary structures), then the inferred transition paths will be similarly biased.

**Essential References Not Discussed:**

The related works are properly cited.

**Experimental Designs Or Analyses:**

The issue has been described in the previous question,  the evaluation experiments need stronger validation. The proposed method is innovative, but its evaluation is insufficient to justify its claims of accuracy, efficiency, and generalizability.

**Methods And Evaluation Criteria:**

The main method introduced in the paper is OM action minimization using generative models (specifically denoising diffusion models and flow matching models) to infer likely transition pathways between molecular configurations. The key steps include training a generative model on equilibrium molecular conformations; using the learned score function as an approximate force field and optimizing paths between two states by minimizing the OM action.

The OM action is based on path probability, but most probable does not equal dynamically correct. The method may find smooth interpolations rather than true dynamical pathways dictated by physical kinetics. Besides, there is no comparison is made with unbiased MD simulations to check if the generated paths follow the correct transition state kinetics.

Another issue is that the optimization heavily depends on parameters controlling diffusion, drift, and friction, but the method does not provide a systematic way to tune these parameters for different molecular systems. For example, the experiments show that lower diffusivity (Figure 2) gives a better estimation of the intrinsic reaction coordinate (minimum energy pathway that connects the saddle point and two minima) while the larger diffusivity is way off the accurate transition path.

A more robust evaluation of TPS methods should include:
Comparison with well-established TPS methods (e.g., umbrella sampling, metadynamics, transition path sampling).
Validation against unbiased MD simulations to assess kinetic accuracy.
Quantitative measures of transition rate, free energy profile, and committor function accuracy.

**Other Comments Or Suggestions:**

N/A

**Other Strengths And Weaknesses:**

Strength: The paper presents a scalable, and computationally efficient approach to transition path sampling using generative models and OM minimization.

Weakness: The paper lacks crucial validation against enhanced sampling methods and transition rate calculations, leaving uncertainty about its physical accuracy. If these issues are addressed in future work, the approach could impact molecular simulations and accelerate the discovery of reaction pathways.

**Questions For Authors:**

1. How does OM optimization compare to well-established enhanced sampling techniques in terms of both efficiency and accuracy? Without this comparison, the claim of “higher efficiency” is weak.

2. How do these transition paths compare quantitatively with TPS or enhanced sampling methods in terms of transition rates and free energy barriers?

3. How does OM optimization perform on systems where the generative model was trained on an entirely different force field or chemical environment? Does it still yield reasonable transition paths?

4. Does this method capture all relevant transition pathways, or does it preferentially generate low-energy interpolations that may miss rare but important transitions?

5. How does OM optimization perform with chemical reaction systems (when bond cleavage/formation is happening)?

**Relation To Broader Scientific Literature:**

The contribution of the paper may benefit the study of transition paths but still lacks confidence due to incomplete validation, limited benchmarking, and potential oversimplification of transition path dynamics. While the idea of leveraging generative models for transition path sampling (TPS) is innovative, the connection between OM action minimization and physically meaningful transition paths remains insufficiently supported. Below, I examine how the paper relates to the broader scientific literature, highlighting where it builds upon previous work and where gaps remain. Due to the high degree of freedom for reaction systems (DOE = 3N-5), the complexity of the potential energy surface can not be solved in this paper. The unique characteristics of the transition path connecting the transition state— which appears as a saddle point in the reaction coordinate projection but as a minimum in other projections—cannot be predicted solely based on the minima.

**Theoretical Claims:**

N/A

---

> ### Author Rebuttal · Authors · 2025-04-01
>
> >Comparison with enhanced sampling baselines
>
> To our knowledge, there’s no widely accepted force field for the $\alpha$-Carbon coarse-graining used for the fast-folding proteins, so benchmarks are challenging. We now benchmark on all-atom alanine dipeptide, a standard test system for TPS. We compare with MCMC (shooting) and metadynamics. See [Table 1r](https://imgur.com/a/iPmIUbp) for accuracy and efficiency results, and [Figure 1r](https://imgur.com/a/SImqo3n) for sampled transition paths.
>
> OM optimization is considerably faster than the baselines. Internal energy barriers were about 10-12 kcal/mol, in line with literature.
> >Quantitative evaluation of transition paths (reaction rate, committor function, free energy profile)
>
> These are known to be challenging to compute accurately for high-dimensional systems. With this in mind, we provide results to show how our OM optimized paths can be used to compute these quantities. We’re happy to include more in-depth results for the camera-ready version.
>
> Free Energy: Our OM-optimized paths can serve as a guide for umbrella sampling, from which we can calculate free energy profiles. We do this for Alanine Dipeptide and achieve the following free energy profile, which closely approximates the barrier of 6-8 kcal/mol obtained from metadynamics: [Figure 2r](https://imgur.com/a/3ci9dXh)
>
> Committor Function/Transition Rates: The Backward Kolmogorov Equation (BKE) provides a principled algorithm [1] to estimate the committor function and transition rates using the paths from OM optimization. See [Figure 3r](https://imgur.com/a/BhLLOKs) for results on the 2D Muller-Brown potential, where we achieve accurate committor and rate estimation, with a margin of error comparable with or better than past works. Note that it is often challenging to even compute the reaction rate within the correct order of magnitude [1,2].
> >Guidance on setting hyperparameters
>
> We now set hyperparameters directly as values with physical units: for example setting $\gamma$ and $D$ to be friction and diffusion coefficients, and $\Delta t$, $L$ from the time horizon and desired fidelity. We provide an example of new paths with varying time horizons in [Figure 4r](https://imgur.com/a/s9DhO3R).
> >Markov State Models (MSM) do not confirm dynamical accuracy
>
> Our reference MSMs are fit on long D.E.Shaw simulations which extensively sample the Boltzmann distribution, including many transition events. Therefore, the reference MSMs are a useful low-dimensional approximation of the underlying dynamics of the unbiased MD simulation, which is also standard practice in past works [3,4].
> We have also directly compared our generated paths against the unbiased MD transition trajectories from D.E.Shaw: [Figure 5r](https://imgur.com/a/7m7Npne)
> >OM Optimization on chemical reactions
>
> We conducted initial experiments using a machine learning force field (MLFF) trained on the Transition-1x dataset. OM interpolation finds transition states on average 1.4eV above the reference transition state (found using Nudged Elastic Band (NEB) calculations with DFT), significantly better than running NEB calculations with the MLFF (1.91eV above) and uses 10x fewer force evaluations. A comprehensive analysis would be a paper in and of itself, so we consider this an interesting future direction.
> >Zero-shot generalization on unseen tetrapeptides
>
> While the force field is the same for all tetrapeptides (see [4]), it is sequence-dependent: the energy landscape is different for held-out proteins compared with training proteins. Thus, our results in Section 5.3 effectively demonstrate generalization to new interaction potentials.
> >Rare events underrepresented in training data
>
> We agree that this is an inherent challenge associated with data-driven methods. However, we have demonstrated experimentally in Section 5.2 that OM optimization is resilient to a degree of data sparsity and still identifies realistic transition paths. As we scale up to larger models like BioEmu [5], we expect resilience to sparsity to get stronger.
> >Preferring low-energy interpolations
>
> We can still achieve diversity in sampled transition paths by leveraging stochasticity of the generative model to get diverse initial guesses (Alg. 2, Sec. G) – converged paths thus tend to find diverse local minima. Further, $\tau_\text{initial}$ can be tuned to vary the diversity of the initial guesses and subsequently produced paths.
> >Solving CV selection problem
>
> We don’t claim to solve the CV selection problem. We just note that our method does not require CV to operate.
> >OM Optimization on completely different force fields or chemical environments
>
> This is an important area for future work, and we believe that using more performant models such as BioEmu [5] as the underlying score estimator would aid in achieving this.
> [1] Hasyim et al. JCP (2022)
> [2] Rotskoff, et al. PMLR (2022)
> [3] Arts et al. JCTC (2023)
> [4] Jing et al. NeurIPS (2024)
> [5] Lewis et al. arxiv: 2024.12.05.626885

---

### Official Review · Reviewer_dGUc · 2025-03-09

**Overall Recommendation:** 4

**Summary:**

The manuscript focuses on transition path sampling (TPS), which involves identifying high-probability paths between two states or points on an energy landscape. The authors combine generative models trained to sample temporally independent states from an energy landscape with the task of transition path sampling. These paths arise from a stochastic differential equation (SDE) constructed from a diffusion or flow-matching model. The authors observe that finding high-likelihood transition paths is equivalent to minimizing the Onsager-Machlup action functional. This connection enables the authors to repurpose pre-trained generative models for TPS. They demonstrate their approach on protein and molecular systems.

**Claims And Evidence:**

The claims made in the manuscript are backed with theoretical analysis and experiments.

**Essential References Not Discussed:**

[1] AlphaFlow could be cited as relevant works.

[1] Jing, B., Berger, B., & Jaakkola, T. (2024). AlphaFold meets flow matching for generating protein ensembles. arXiv preprint arXiv:2402.04845.

**Experimental Designs Or Analyses:**

The results are interesting and demonstrate strong performance. However, I believe comparisons against existing methods or simpler baselines are lacking. For example, given access to both the score and probability, you could try a simple Markov Chain Monte Carlo (MCMC) algorithm like Metropolis-Adjusted Langevin Algorithm (MALA) or Hamiltonian Monte Carlo (HMC)? Similar works, like AlphaFlow, are relevant to molecular dynamics, and there exist diffusion and flow-matching generative models for proteins that could be integrated within your framework. This suggestion is to enhance comparisons with existing methods; if you know of any others that would be appropriate, that would be welcome.

**Methods And Evaluation Criteria:**

The method is grounded in strong theoretical results.

**Other Comments Or Suggestions:**

- > However, these approaches rely on highly specialized training procedures and fail to exploit the growing quantity of atomistic simulation and structural data

It would be helpful to provide an example of these specialized training procedures or at least elaborate on why this is a drawback of existing approaches.

**Other Strengths And Weaknesses:**

I found the introduction and background sections well written and very informative. The method is theoretically grounded, and its explanation is clear.

The primary drawback is the lack of extensive comparisons in the experimental sections.

**Questions For Authors:**

- I would like to confirm that the reason it is zero-shot is that the Onsager-Machlup (OM) minimization only occurs at inference? There is no need for a fine-tuning stage?
Repurpose pre-trained generative models for TPS in a zero-shot fashion.

- Have you attempted using the OM for generative tasks in a manner similar to your current approach, but starting with a noise representation?

- It would be beneficial to clarify the differences between your work and [2].

It is mentioned in the limitations, but what are the advantages of your method compared to this one (if I understand correctly, they need to train or fine-tune)?

[2] Du, Y., Plainer, M., Brekelmans, R., Duan, C., NoÅLe, F., Gomes, C. P., Aspuru-Guzik, A., and Neklyudov, K. Doob’s Lagrangian: A sample-efficient variational approach to transition path sampling. arXiv preprint arXiv:2410.07974, 2024.

- In diffusion and flow-matching models, the score of the vector field is usually parameterized by a time variable that interpolates between noise and data. Since, in your case, this initial point is not necessarily noise, which time parameter do you use? I assume the time variable from Eq. 11 will not be used to compute the score $s_\theta(x,\tau)$, especially since $\tau$ needs to fall within the interval originally used to train the model.

**Relation To Broader Scientific Literature:**

The authors cleverly combine existing theoretical findings and adjust flow matching (FM) to their framework. Their work could be used for various applications that already use diffusion models or flow-matching.

**Theoretical Claims:**

The authors state that the SDE arising from Denoising Diffusion Probabilistic Models (DDPM) is a natural candidate for TPS, but not necessarily a good one. Was this experimentally tested? If so, it may be useful to include a reference to the experiment.

>While the denoising (i.e., sampling) process of a DDPM (see Appendix B.1) may appear to be a natural candidate, a closer inspection reveals that it is unsuitable, as it optimizes for different likelihoods at different points along the trajectory. A large portion of the denoising trajectory thus has low likelihood under the data distribution. Therefore, we need to consider an alternative approach.

---

> ### Author Rebuttal · Authors · 2025-04-01
>
> >Comparison against baselines and incorporation of other generative models
>
> Please see our response to reviewer bRih, in which we compare our OM optimization approach on alanine dipeptide with two traditional approaches for transition path sampling: Markov Chain Monte Carlo (MCMC)  and metadynamics. We find that OM optimization is several orders of magnitude faster than these approaches and finds transition paths with energy barriers that agree with the reported literature values. We agree that it would also be very interesting to incorporate AlphaFlow and other recent generative models into our OM framework. We have done some preliminary work with AlphaFlow, but found it challenging to work with due to its reliance on pre-trained OpenFold/ESM models, which are extremely memory-intensive. We are actively working to address these challenges in future work.
>
> >Unsuitability of denoising diffusion SDE for transition path sampling
>
> The full direct denoising process starts with maximum time conditioning ($t=T$) at Gaussian noise (i.e. the target of the learned vector field $\epsilon_\theta(x, T)$ is the score of a pure Gaussian). Therefore, optimizing paths to have high likelihood under the denoising SDE would provide a “force field” near the start of the trajectory that just forces it to the origin. Thus the denoising SDE is ill-posed for our setting of TPS, since we want our SDE to produce paths which are high-likelihood under the data distribution at all times, not just at the final denoising step.
>
> >Example of specialized training procedures in prior work and their drawbacks.
>
> Specialized training procedures in past works include solving a Schrodinger Bridge problem via Stochastic Optimal Control, reinforcement learning, or differentiable simulations (see our submission for citations). These training procedures are expensive because they involve running MD simulations during training, and most trajectories fail to reach the target state, yielding sparse rewards and high cost. A simulation-free approach that guarantees endpoint constraints was introduced in [1]. However, all of these techniques require a training process which is **unique to transition path sampling and must be repeated for every new system of interest (since they do not utilize any training data)**, which is expensive. Meanwhile, our approach indirectly leverages the data and compute used to train atomistic generative models (which can be used for many tasks beyond TPS, notably conformational sampling), using a lightweight test-time OM action minimization procedure.
>
> >Comparison to Doobs Lagrangian
>
> Doobs Lagrangian [1] is a data-free method, meaning it relies only on querying the underlying energy/force function, and requires a bespoke training procedure for every molecular system of interest in order to learn the optimal bias potential for TPS. Meanwhile, our approach leverages pretrained generative models off-the-shelf, with no specialized training procedure for TPS, and can be used across chemical space (barring excessive distribution shift from the generative model’s training data), as we demonstrate in our “Generalization to Unseen Tetrapeptides” experiments.
>
> >Using OM optimization for generative tasks
>
> Optimizing the denoising process directly could be feasible, and indeed falls into the OM optimization framework (essentially by changing $\tau_\text{opt}$ between endpoints, similar to $\tau$ in the denoising process). However, since our paper’s setting focuses on optimizing trajectories that are high likelihood under the data distribution throughout the entire trajectory (rather than just at the endpoint), we have not run such experiments here. This is an interesting idea for future work!
>
> >Zero-shot property of OM optimization.
>
> OM optimization only occurs at test-time, and is a gradient descent procedure over the transition path, not over the generative model weights. The generative model weights are completely fixed throughout our procedure, which is why we call it “zero-shot”.
>
> >What time parameter is used in OM optimization?
>
> Due to space constraints, please see our response to reviewer jnQt, who asked a similar question.
>
> [1] Du et al. NeurIPS (2024)

---

> > ### Comment · Reviewer_dGUc · 2025-04-04
> >
> > I would like to thank the authors for their response. I don't have additional questions at the moment.

---

> > > ### Author Response · Authors · 2025-04-05
> > >
> > > We thank the reviewer for their helpful comments and for increasing their score.

---

### Official Review · Reviewer_jnQt · 2025-03-11

**Overall Recommendation:** 4

**Summary:**

The paper proposes a way of using a score-based or a flow based generative model trained to generate molecular configurations to generate transition paths between meta-stable configurations. The paper proposes so be drawing a relation between what would be a limit SDE corresponding to noising and denoising process of DDPM between two adjacent times and the original SDE coming from the physics of the molecular configurations and its energy potential. This relationship allows to adapt the minimization of the Onsager-Machlup functional to the score from the generative model. The paper then proceeds to a numerical evaluation in diverse settings where they compare the proposed approach to state of the art and show that it is able to sample paths that look plausible from several different metrics and comparable with much more compute demanding simulations.

**Claims And Evidence:**

Yes

**Essential References Not Discussed:**

N/A

**Experimental Designs Or Analyses:**

Yes, the experiments are pertinent and show the value of the proposed method. I still have an issue with the lack of discussion of the optimal diffusion times. I think it would be interesting to see for example in the toy example what is the tradeoff between diffusion time $\tau_{opt}$ and the distance between the wells. As I see it, having more distant metastables states would lead to an augmentation of the $\tau_{opt}$ used.

**Methods And Evaluation Criteria:**

The proposed datasets are pertinent to the problem at hand.

**Other Comments Or Suggestions:**

N/A

**Other Strengths And Weaknesses:**

The paper present an original idea for using generative models to sample joint transition paths, which open new research venues. The main weakness of the paper is in my opinion the lack of further investigation around the score based model and it's trade-offs and also the fact that the proposed method is rather adhoc as said in the limitations sections.

**Questions For Authors:**

1. Could the authors further comment figure 7? It seems that in this case flow matching is actually able to sample configurations around the transition are but the actual OM minimization lead to some erroneous trajectories.
2. Why not simply use Langevin dynamics with the learned score instead of the noising denoising limiting equation? (11)

**Relation To Broader Scientific Literature:**

The paper proposes a method relying on pre-trained generative models for the molecular configuration for sampling transition paths in a much faster way than the other proposed methods in the literature, such as learning the control drift through reinforcement learning or direct simulation. Although it is has less theoretical guarantees, it can still be a valuable addition.

**Theoretical Claims:**

yes

---

> ### Author Rebuttal · Authors · 2025-04-01
>
> > Discussion of optimal diffusion time
>
> A couple of ways that $\tau_\text{opt}$ is chosen:
>
> 1. For diffusion models, a small nonzero $\tau$ usually works better than $\tau=0$. Theorem B.1 highlights why, noting $\bar \alpha_0 = 1$. Scores closer to the data are weighted lower in a standard DDPM training process. We have run some cosine similarity tests with the true force over Muller-Brown and Alanine Dipeptide to inform the optimal $\tau_\text{opt}$ used in these settings (presented in [Figure 8r](https://imgur.com/a/UDyWcxK)) and we used the reported optimal times for fast folders [1].
>
> 2. Even if $s_\theta(x, \tau_\text{opt})$ matches the true score well for small enough $\tau_\text{opt}$, this vector field can be difficult to optimize over. We have found at times that annealing over the conditioning time $\tau_\text{opt}$ helps the OM optimization: start at a value of $\tau_\text{opt}$ closer to noise, where $s_\theta(x, \tau_\text{opt})$ is generally a smoother vector field, then gradually over the optimization anneal $\tau_\text{opt}$ closer to data. This then motivates us deriving eq. (11) for any latent time, rather than just close to the data distribution.
>
> [1] Arts et al. JCTC (2023)
>
> > “Ad-hoc” nature of the method
>
> Please see our response to Reviewer bRih, who also raised a similar concern that the hyperparameters controlling the relative contributions of the path, force, and diffusion terms in the OM action are set in an ad-hoc manner. We have refined our method such that these parameters are now directly interpretable as constants with physical units, rather than arbitrary hyperparameters. This makes choosing these hyperparameters intuitive and constrains the hyperparameter search space. More generally, as presented in the Appendix, our method is rigorously derived from well-known action-minimization principles from statistical mechanics, and the equivalence between the learned score function from generative models and the true force field has a clear theoretical justification in the limit of a large, expressive model trained on comprehensive data.
>
> > Erroneous trajectories in flow matching (Figure 7):
>
> There are a couple factors that make optimized trajectories (even at 0 temperature) look like they’re not following the data distribution:
>
> 1. There were fewer plotted samples in order to maintain figure simplicity, but as a result what is not displayed is the fact that the learned distribution does cover a wider band in the transition path, and the converged path is finding the shortest way through this band (hugging the inner wall).
>
> 2. The trained flow matching model is not perfect, and has some erroneous local minimum at the center, which gets picked up more at higher temperatures and thus requires a more conservative (lower) $dt$ parameter.
> If we increase waypoints and are more careful with the optimization, we can obtain more reasonable trajectories even under a potentially erroneous flow matching model, see the now-provided [Figure 9r](https://imgur.com/a/xxpKHrI).
>
> > Using Langevin dynamics instead of the noising-denoising limiting equation
>
> Eq. (11) is essentially Langevin dynamics with a learned score at a fixed time-conditioning (note the reasoning works at any time in the noising process, where the vector field is the score of a noised distribution rather than the data distribution). Meanwhile, the standard Langevin dynamics used for sampling/denoising varies the score from $s_\theta(x, T)$ to $s_\theta(x, 0)$ throughout the trajectory.
>
> The motivation for using the fixed-time, noising-denoising limiting equation is the following: our desiderata for the SDE is that the entire trajectory generated by the SDE remains high-likelihood under the data distribution. For variable-time Langevin dynamics, this desiderata is not satisfied, as only the converged/steady-state limit of the dynamics produces data samples. Meanwhile, the noising/denoising equation satisfies this criterion, and the limit process is a valid SDE to which we can apply OM optimization.

---

### Official Review · Reviewer_Z19m · 2025-03-14

**Overall Recommendation:** 4

**Summary:**

The paper introduces Onsager-Machlup (OM) optimization to sample transition paths, claiming three advantages: efficiency, scalability, and flexibility. OM optimization approach produce transition paths in pre-trained generative models, where the core idea is interpreting candidate paths as the denoise-noise SDE allowing tractable computation. Experiments on 2D Muller-Brown potential and fast-folding coarse proteins show that the method produces realistic paths and is also generalized to unseen tetra-peptides.

## Update after rebuttal

I have confirmed the author's response on the scalability and efficiency concern of the proposed method, and raised the score accordingly.

**Claims And Evidence:**

The paper’s main claim is supported by theoretical arguments (section 3) and experiments (section 5). Specially for the efficiency aspect, transition paths are generated using pre-trained generative models, resulting valid metrics compared to MSM in smaller cost.

**Essential References Not Discussed:**

It seems all of related works have been comprehensively discussed, with details in Appendix A.

**Experimental Designs Or Analyses:**

The experimental design and analyses are concrete and appropriate.

- 2D Muller brown: well-known synthetic testbed for transition path studies
- Fast-folding protein systems: evaluation and analysis with MSMs
- Tetra-peptides: transition path generation for unseen data, with evaluation and analysis with MSMs

**Methods And Evaluation Criteria:**

The proposed method and evaluation criteria are suited for the problem of transition path sampling, but additional evaluation criteria may strengthen the author’s claim

- 2D Muller brown: qualitative evaluation is given in the main paper, and additional contents in the appendix. A distribution plot of the transition state or the highest energy in 2D Muller brown would clearly show that the OM optimization works well.
- Fast-folding coarse-grained proteins, tetra-peptides: evaluation based on MSM following MDGen is well done

**Other Comments Or Suggestions:**

I do not have any other comments

**Other Strengths And Weaknesses:**

**Strengths**

1. Originality

Combining OM optimization with pre-trained generative model seems genuinely enough.

2. Presentation

The paper is well-written, logically structured making it easy to follow!

3. Extensive experiments

Along with synthetic systems, fast-folding proteins and tetra-peptides experiments are well done and validate the authors claim


**Weaknesses**

1. (Minor) diversity for transition paths

While “diversity” is not rigorously quantified (no explicit diversity metric is given), the authors do generate multiple paths and report that many are unique and all have non-zero probability under the reference MSM.

2. Comparison with MDGen [1]

MDGen is a generative model targeting molecular systems with multiple downstream tasks, with transition path sampling being one of them. Like the evaluation in this paper, it also uses MSMs for evaluation. The generalization task for tetra-peptides seems quite similar to that of MDGens. Is there any comparison between MDGen with OM optimization, e.g., OM is efficient to MDGen in terms of GPU hours?

[1] Generative Modeling of Molecular Dynamics Trajectories, NIPS 2024

**Questions For Authors:**

1. Scalability

The authors highlighted that the proposed method is advantageous in scalability, proteins and tetra-peptides are done by coarse-graining leading to less than hundred of atoms. Since MDGen[1] models where tetra-peptides are modeled in all atoms, I am confused what the authors imply by scalability. Could the authors provide some details about scalability compared to prior works?

[1] Generative Modeling of Molecular Dynamics Trajectories, NIPS 2024

**Relation To Broader Scientific Literature:**

The key contributions of this paper is related to ‘transition path sampling’, where traditional methods have been struggling due to extensive computation. The proposed method seems efficient compared to prior works (however, please check weakness 2).

**Theoretical Claims:**

I’ve gone through the core theoretical claim, using Onsager-Machlup action for paths under SDE and can be used to produce realistic transition paths with pre-trained generative models.

---

> ### Author Rebuttal · Authors · 2025-04-01
>
> > Scalability and efficiency comparison to previous works
>
> To address the concern about only including results on coarse-grained tetrapeptides, we also present OM optimization results on all-atom tetrapeptides, which contain up to 56 atoms, in [Figure 6r](https://imgur.com/a/naQLWDy). We obtain competitive results on the Markov State Model metrics, similar to coarse-grained tetrapeptides in Figure 4 of the paper. We plan to replace Figure 4 with this all-atom result for the camera-ready version of the paper.
>
> Regarding efficiency relative to past works, MDGen [1] requires a TPS-specific training procedure of about 460 GPU-hours, followed by inference on the order of a few seconds per tetrapeptide path (numbers obtained via direct correspondence with the authors). Meanwhile, we utilize pretrained generative models and perform OM optimization using the learned score function, which requires approximately 30 seconds per tetrapeptide path.
>
> To account for the increased memory footprint of larger systems, we use a simple path batching technique, which splits up the discretized path into mini batches (e.g split a large, 5,000-point path into mini-batches of size 100). The batches can be optimized either sequentially or trivially parallelized with multiple GPUs at each step of OM optimization, making our method scalable.
>
> At a broader level, our method is scalable in the sense that we expect continual performance improvements as the underlying atomistic generative models scale up and learn better score functions. This also applies to the growth in size and quality of underlying training datasets (including incorporating experimental data, etc.). We believe that our approach, which requires no TPS-specific training procedure, aligns well with the growing trend of leveraging large-scale, well-tested, general-purpose generative models—a direction already standard in language, speech, and image modeling communities. As high-quality generative models become increasingly available, this synergy will position our method more favorably than existing TPS approaches, which lack such compatibility.
>
> [1] Jing et al. NeurIPS (2024)
>
> > Transition state distribution for 2D Muller Brown
>
> We show the distribution of sampled transition (i.e highest-energy) states for 2D Muller Brown in [Figure 7r](https://imgur.com/a/CxPwszr). While OM optimization does not capture the full transition state ensemble, this can easily be obtained by initiating MD simulations along the sampled path.
>
> > Diversity metric
>
> We did not report an explicit diversity metric because to our knowledge, there is no such agreed upon metric in the TPS literature. While not directly a diversity measure, reported distributional distances using the Jensen-Shannon Divergence (JSD) between the MSM state distribution visited by the true and reference transition paths (see Fig. 3d and 4) serve as a proxy for capturing the appropriate path diversity. If the diversity was under- or over-estimated, this would be reflected in the JSD.

---

### Decision · Program_Chairs · 2025-05-01

**Decision:**

Accept (poster)

**Comment:**

The author present an application of the Onsager-Machlup action minimization functional to sample transition paths in pretrained molecular ensemble generation models, e.g. pre-trained Boltzmann Generators or emulators. While the approach is conceptually simple and well-known in the molecular simulation community (work by Vanden-Eijden and others is discussed in the paper), however, their practicality have been limited by computational demands in its implementation. The application to pre-trained generative models is as far the reviewers and I am concerned, new. However, when applied this setting the sampled transition paths will only be as accurate as the underlying surrogate allows. The reviewers are positive overall, however, their enthusiasm appears to hinge to some degree of the promise of scaling these approaches to larger and more challenging systems than the ones explored the in the paper. The authors acknowledged this and promises a more extensive set of systems in their final paper.